# Prognostic value of the right ventricular ejection fraction using three-dimensional echocardiography: Systematic review and meta-analysis

Tetsuji Kitano[1]*, Yosuke Nabeshima[1], Yasufumi Nagata[1], Masaaki Takeuchi[2]

**1** Second Department of Internal Medicine, University of Occupational and Environmental Health, School of Medicine, Kitakyushu, Japan, **2** Department of Laboratory and Transfusion Medicine, University of Occupational and Environmental Health Hospital, Kitakyushu, Japan

\* syuukyuu1986@gmail.com

## Abstract

### Aims

Three-dimensional echocardiography (3DE) is a robust method for measuring the right ventricular (RV) ejection fraction (EF), which is closely associated with outcomes. We performed a systematic review and meta-analysis (1) to examine the prognostic value of RVEF and (2) to compare its prognostic value with that of left ventricular (LV) EF and LV global longitudinal strain (GLS). We also performed individual patient data analysis to validate the results.

### Methods and results

We searched articles reporting the prognostic value of RVEF. Hazard ratios (HR) were rescaled using the within-study standard deviation (SD). To compare predictive values of RVEF and LVEF or LVGLS, the ratio of HR related to a 1-SD reduction of RVEF versus LVEF or LVGLS was calculated. Pooled HR of RVEF and pooled ratio of HR were analyzed in a random-effects model. Fifteen articles with 3,228 subjects were included. Pooled HR of a 1-SD reduction of RVEF was 2.54 (95% confidence interval (CI): 2.15–3.00). In subgroup analysis, RVEF was significantly associated with outcome in pulmonary arterial hypertension (PAH) (HR: 2.79, 95% CI: 2.04–3.82) and cardiovascular (CV) diseases (HR: 2.23, 95%CI: 1.76–2.83). In studies reporting HRs for both RVEF and LVEF or RVEF and LVGLS in the same cohort, RVEF had 1.8-fold greater prognostic power per 1-SD reduction than LVEF (ratio of HR: 1.81, 95%CI: 1.20–2.71), but had predictive value similar to that of LVGLS (ratio of HR: 1.10, 95%CI: 0.91–1.31) and to LVEF in patients with reduced LVEF (ratio of HR: 1.34, 95%CI: 0.94–1.91). In individual patient data analysis (n = 1,142), RVEF < 45% was significantly associated with worse CV outcome (HR: 4.95, 95% CI: 3.66–6.70), even in patients with reduced or preserved LVEF.

**Data Availability Statement:** All relevant data are within the manuscript and its Supporting Information files.

**Funding:** The authors received no specific funding for this work.

**Competing interests:** The authors have declared that no competing interests exist.

## Conclusions

The findings of this meta-analysis highlight and support the use of RVEF assessed by 3DE to predict CV outcomes in routine clinical practice in patients with CV diseases and in those with PAH.

## Introduction

Evaluation of right ventricular (RV) systolic function is important to diagnose RV dysfunction and to predict outcome [1–6]. RV function may be a better predictor of mortality than left ventricular (LV) indices in myocardial infarction and chronic heart failure [7, 8]. The most commonly used parameters for RV systolic function are tricuspid annular plane systolic excursion (TAPSE), tricuspid annular peak systolic velocity (s'), RV fractional area change (RVFAC), and RV free-wall longitudinal strain (RVFWLS). Since these parameters represent functioning of either a limited part of the right ventricle or one cut-plane of the right ventricle, they may not reflect global RV function. Given the complex morphology and dynamic motion of the right ventricle, RV ejection fraction (RVEF) is a more reliable and robust marker of RV function. Cardiac magnetic resonance (CMR) has been the gold standard for measuring RVEF because of its high temporal and spatial resolution, accuracy, and reproducibility [9, 10]. However, CMR is expensive, time-consuming, limited in availability, and is not suitable for all patients. On the other hand, three-dimensional echocardiography (3DE) is more versatile, repeatable, and less costly. Recently, advances in ultrasound equipment and software in 3DE have also improved further, so that 3DE has become a potential alternative for evaluation of RVEF in clinical practice [11, 12].

The prognostic value of RVEF assessed by 3DE has been reported in several previous studies, and its usefulness has been demonstrated not only in right heart disease, such as pulmonary arterial hypertension (PAH), but also in left heart disease [1–3, 6]. Just recently, Sayour et al. conducted a systematic review and meta-analysis of the association of 3DE-derived RVEF with adverse cardiopulmonary outcomes and compared the prognostic value of RVEF to TAPSE, RVFAC, and RVFWLS [13]. However, RVEF has not been compared to LV parameters such as LVEF and LV global longitudinal strain (LVGLS), which are the most important parameters in patients with cardiovascular (CV) disease. Accordingly, we conducted this systematic review and meta-analysis (1) to examine the prognostic value of 3DE-derived RVEF and (2) to compare its prognostic value with that of LVEF and LVGLS in the same subjects. We also performed individual patient data analysis using publications from our laboratory to validate results.

## Materials and methods

### Search strategy and selection of articles

This systematic review and meta-analysis were performed in accordance with PRISMA (Preferred Reporting Items for Systematic reviews and Meta-Analysis) guidelines [14]. A literature search was conducted in three electronic databases (PubMed, Embase, and Scopus) using the keywords "right ventricular ejection fraction," "three-dimensional echocardiography," "prognosis," and related phrases (S1 Table). Inclusion criteria were (1) studies with adult subjects, (2) full-text articles in peer-reviewed journals, (3) studies reporting the mean and standard deviation (SD) of RVEF by 3DE, and (4) studies reporting hazard ratios (HRs) for the

association between 3DE RVEF and outcomes. Exclusion criteria were (1) studies including children, (2) studies including non-human subjects, and (3) case reports, conference abstracts, reviews, editorials, or comments. Two authors (TK, YN) reviewed these search results separately and included only articles that met the aforementioned criteria. This study was prospectively registered in the PROSPERO database (CRD42022300188).

## Data extraction

The year of publication, journal, number of subjects, mean age, gender, background disease, ultrasound vendor, software used for the analysis, primary endpoint, follow-up duration, number of events, mean and SD of RVEF, and HR and its 95% confidence interval (CI) for the association RVEF per unit change and outcomes were retrieved. In addition, if available, means and standard deviations of LVEF and/or LVGLS, and HRs and 95% CIs for the association between LVEF and/or LVGLS and outcomes were extracted. When a study reported more than one endpoint, we selected the hardest one. If we found several publications using a similar group of patients from the same authors, we selected the publication with the largest patient cohort.

## Validation analysis

To validate results of this meta-analysis, a validation analysis was conducted using individual patient data from three of four previous studies in our laboratory [1, 2, 15]. One study was excluded because the majority of patients overlapped and the number of study populations was smaller than the others.

## Statistical analysis

Statistical analysis was performed using R version 4.2.1 (The R Foundation for Statistical Computing, Vienna, Austria). Continuous data were expressed as the mean ± SD, and categorical variables were presented as absolute numbers or percentages. To allow comparisons among studies, HRs and 95% CIs were re-scaled by the within-study SD for each parameter and represented changes standardized by the absolute value of each parameter [13, 16]. By re-scaling this way, the hazard for a 1-SD reduction in the parameters could be directly compared. Furthermore, to enable comparisons between RVEF and LVEF or between RVEF and LVGLS, ratios of HR related a 1-SD reduction in RVEF versus LVEF or LVGLS were calculated. A ratio > 1 indicates that a 1-SD reduction in RVEF is associated with a greater hazard increase relative to a 1-SD reduction in other parameters (LVEF or LVGLS). A random effects model was used for analysis and results are displayed in a forest plot. Publication bias was assessed using funnel plots and Egger's test [17]. The Cochran Q test and the inconsistency factor ($I^2$) were used to assess heterogeneity among studies. Meta-regression analysis was performed to determine whether publication year, type of endpoint, and follow-up duration affect HR of RVEF. Subgroup analysis was conducted according to the background disease [PAH or CV disease] and modality (2DE or 3DE). Quality assessment of these studies was conducted using the quality assessment tools of Downs and Black [18]. A two-sided P value <0.05 was used to determine statistical significance, whereas a P value <0.1 was used for Egger's test.

Regarding validation analysis, we performed Kaplan-Meyer survival analysis among four groups according to 1) LVEF < 45% or ≥ 45% and RVEF < 45% or ≥ 45%, and 2) LVGLS < 16% or ≥ 16% and RVEF < 45% or ≥ 45%. Next, uni- and multi-variable Cox proportional hazards regression analyses were performed. Subgroup analyses were also conducted in patients with LVEF < 45% or > 45% and LVGLS < 16% or > 16%. Finally, sequential Cox

regression analysis was conducted to determine incremental value of RVEF over LVEF (LVGLS) and clinical parameters.

## Results

A PRISMA flowchart depicting the selection process is shown in Fig 1. From 1,592 articles retrieved from three databases, 15 articles with 3,228 subjects were finally included in this meta-analysis [1–3, 6, 15, 19–28]. Table 1 shows characteristics of study subjects. Publication years ranged from 2016 to 2022. Four studies included only PAH and five studies analyzed RVEF in CV diseases (Table 2). Six other studies examined specific types of patients. The primary endpoint was death (any cause, cardiac or cardiopulmonary death) in five studies and 10 other studies used a composite endpoint [all cause death, cardiac death, heart failure (HF) hospitalization, ventricular tachyarrhythmia, non-fatal myocardial infarction, PAH-related hospitalization, or PAH-related intervention]. Follow-up duration ranged from 3 months to 80.4 months. Of the 15 articles, LVEF was reported in 12, including 2,867 patients, and LVGLS was reported in 9 articles, including 2,214 patients. S2 and S3 Tables summarize reasons for full-text exclusion and the study quality analysis.

### Prognostic value of RVEF

**All CV diseases, including pulmonary arterial hypertension.** Forest plots for HR of RVEF are shown in Fig 2. In 15 studies, a 1-SD reduction of RVEF was associated with a 2.54-fold (95% CI: 2.14–3.00, p<0.001) increase in risk of death or cardiovascular events. There was moderate heterogeneity among the included studies ($I^2$: 66%, $\tau^2$: 0.052). The funnel plot shows asymmetry and Egger's test confirmed publication bias (p = 0.028). There were two outliers [3, 27]. If we excluded these two outliers, our pooled study estimate did not change

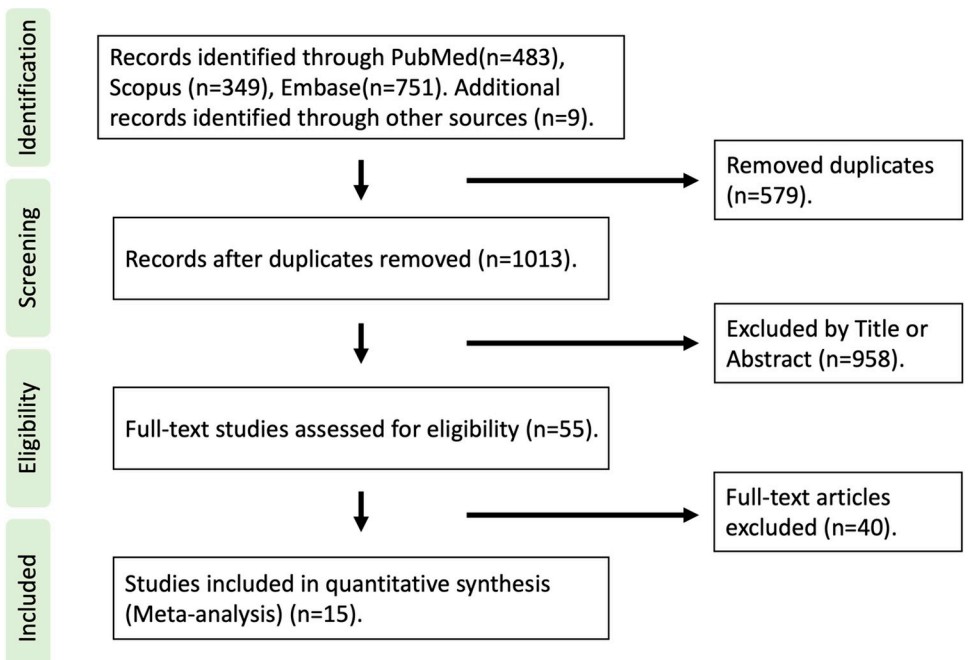

**Fig 1. Preferred Reporting Items for Systematic reviews and Meta-Analysis (PRISMA) flow chart of study selection.**

**Table 1. Characteristics of study subjects.**

| First author | Year | Number | Age (mean± SD) | Male (%) | Disease type | Vendor | End point | Follow-up (months) | Events (%) |
|---|---|---|---|---|---|---|---|---|---|
| Murata | 2016 | 86 | 50±17 | 23 | PAH | TomTec | Composite | 14.1 | 19 |
| Nagata | 2017 | 446 | 66±16 | 259 | various CVD | TomTec | CD | 49.2 | 38 |
| Moceri | 2018 | 104 | 66 | 46 | PAH | TomTec | CPD | 6.6 | 16 |
| Surkova | 2019 | 394 | 56 | 258 | various CVD | TomTec | ACD | 44.4 | 56 |
| Li | 2020 | 54 | 54±16 | NA | PAH | TomTec | Composite | 28 | 20 |
| Muraru | 2020 | 412 | 57 | 268 | various CVD | TomTec | Composite | 44.5 | 59 |
| Li | 2021 | 203 | 49±15 | 57 | PAH | Philips | Composite | 20.9 | 87 |
| Meng | 2021 | 81 | 62 | 53 | HFpEF | TomTec | Composite | 17 | 39 |
| Nabeshima | 2021 | 367 | 77±10 | 168 | AS | TomTec | Composite | 26.7 | 57 |
| Surkova | 2021 | 292 | 59±17 | 203 | various CVD | TomTec | Composite | 80.4 | 107 |
| Tolvaj | 2021 | 174 | 62±14 | 126 | HFrEF, HTx, MVR | TomTec | ACD | 24 | 24 |
| Vijiiac | 2021 | 50 | 61±14 | 34 | DCM | TomTec | Composite | 16 | 29 |
| Zhang | 2021 | 128 | 61±13 | 61 | COVID-19 | Philips | ACD | 3 | 18 |
| Kitano | 2022 | 341 | 67 | 226 | various CVD | TomTec | Composite | 19.8 | 49 |
| Shen | 2022 | 96 | 50 | 57 | B-cell lymphoma | TomTec | Composite | 73.2 | 18 |

Composite endpoint: all cause death, cardiac death, heart failure hospitalization, ventricular tachyarrhythmia, non-fatal myocardial infarction, PAH-related hospitalization, or PAH-related intervention.

ACD, all cause death; AS, aortic stenosis; CD, cardiac death; COVID-19, coronavirus infectious disease, emerged in 2019; CPD, cardiopulmonary death; CVD, cardiovascular disease; DCM, dilatated cardiomyopathy; HFpEF, heart failure with preserved ejection fraction; HFrEF, heart failure with reduced ejection fraction; HTx, heart transplantation; MVR, mitral valve replacement; NA, not available; PAH, pulmonary artery hypertension; SD, standard deviation.

**Table 2. Main clinical findings of studies with PAH and CV diseases.**

**A: PAH**

| First author | Year | Number | Feasibility | RVEF (%) | LVEF (%) | LVGLS (%) | Primary outcome | Follow-up (months) | HR by RVEF | HR by LVEF | HR by LVGLS |
|---|---|---|---|---|---|---|---|---|---|---|---|
| Murata | 2016 | 86 | 0.87 | 43±12 | 69±8 | NA | Composite | 14.1 | 0.92 (0.88–0.96) | 1.05 (0.99–1.12) | NA |
| Moceri | 2018 | 104 | 0.92 | 35.6±9.7 | 69.1±8.5 | -8.4±3.6 | CPD | 6.6 | NA | NA | NA |
| Li | 2020 | 54 | 0.81 | 38±10 | NA | -15.4±6.8 | Composite | 28 | 0.92 (0.87–0.98) | NA | 1.27 (1.15–1.41) |
| Li | 2021 | 203 | 0.89 | 37.5±9.1 | NA | -18.6±5.5 | Composite | 20.9 | 0.90 (0.84–0.95) | NA | NA |
| **B: CV diseases** | | | | | | | | | | | |
| Nagata | 2017 | 446 | 0.98 | 50±10 | 48±13 | -12.8±4.1 | CD | 49.2 | 0.90 (0.88–0.93) | 0.94 (0.92–0.96) | 1.23 (1.13–1.34) |
| Surkova | 2019 | 394 | 0.85 | 47±8.1 | 53.6±13.3 | -16.6±5.3 | ACD | 44.4 | 0.90 (0.88–0.92) | 0.96 (0.94–0.97) | 1.14 (1.07–1.21) |
| Muraru | 2020 | 412 | 0.63 | 47±7.4 | 53.7±13.3 | NA | Composite | 44.5 | 0.91 (0.81–0.96) | 0.93 (0.81–0.97) | NA |
| Surkova | 2021 | 292 | 0.85 | 46.5±9.2 | 49.5±14.3 | -14.7±5.1 | Composite | 80.4 | 0.94 (0.93–0.96) | 0.96(0.95–0.97) | 1.11 (1.07–1.16) |
| Kitano | 2022 | 341 | 0.98 | 46.7±11.1 | 39.7±16.3 | -15.2±4.7 | Composite | 19.8 | 0.93 (0.91–0.96) | 0.96 (0.94–0.98) | 1.12 (1.05–1.19) |

Data are expressed as number or means ± standard deviations.

Composite endpoint: all cause death, cardiac death, heart failure hospitalization, ventricular tachyarrhythmia, non-fatal myocardial infarction, PAH-related hospitalization, or PAH-related intervention.

CV, cardiovascular; HR, hazard ratio; LVEF, left ventricular ejection fraction; LVGLS, left ventricular global longitudinal strain; PAH, pulmonary arterial hypertension; RVEF, right ventricular ejection fraction. Other abbreviations are the same as in Table 1.

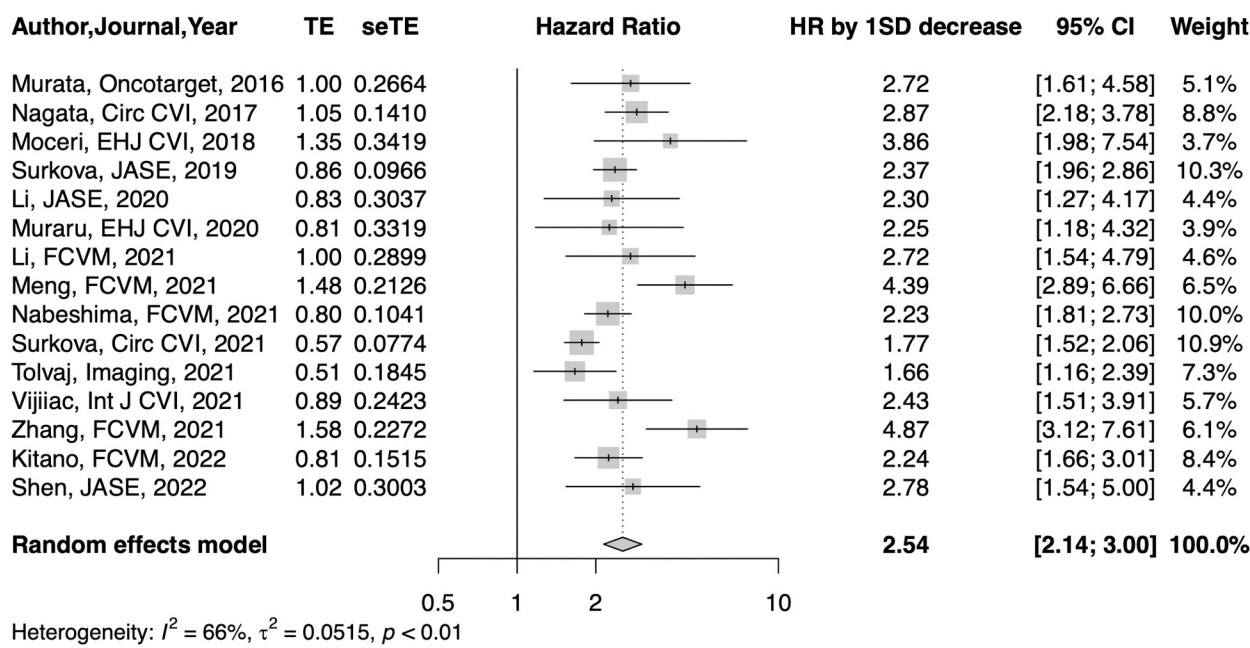

**Fig 2. Forest plots of HR per 1-SD reduction of RVEF.** CI, confidence interval; HR, hazard ratio; SD, standard deviation; se, standard error; TE, test of estimate.

significantly (HR: 2.49, 95% CI: 2.17–2.86, p<0.001). Trim-and-fill analysis added six studies, and the calculated effect size was 2.11 (95% CI: 1.72–2.58, p<0.001). Influence analysis revealed that there were no influenced cases. Leave-one-out analysis showed that omitting Surkova's study [25] decreased $I^2$ to 49%. Meta-regression analysis revealed that publication year or type of endpoint (death or composite) did not affect HR of RVEF. However, total follow up duration was negatively correlated with HR of RVEF (estimate: -0.0065, z value: -2.105, p = 0.035).

**Pulmonary arterial hypertension.** In four studies that dealt with PAH, the pooled estimate was 2.79 (95% CI: 2.04–3.82, p = 0.002) (S1 Fig).

**CV diseases.** In five studies investigating CV diseases, a 1-SD reduction of RVEF was associated with a 2.23-fold (95% CI: 1.76–2.83, p<0.001) increase in risk for adverse outcomes (S1 Fig).

## Prognostic value of RVEF compared to LVEF

In 12 studies that provided both HR of LVEF and RVEF, a 1-SD reduction in LVEF (HR: 1.43, 95% CI: 1.10–1.86, p = 0.011) and RVEF (HR: 2.50, 95% CI: 2.05–3.05) was significantly associated with adverse outcomes (S2 Fig). However, the HR per 1-SD reduction for RVEF was 1.81 times larger (95% CI: 1.20–2.71, p = 0.008) than that for LVEF, with high heterogeneity ($I^2$: 83.5%, $\tau^2$: 0.223) (Fig 3). LVEF was measured with two-dimensional echocardiography (2DE) in three studies, and with 3DE in nine studies. HR for RVEF was 4.36 times higher (95% CI: 2.09–9.08, p<0.001) than that for LVEF with no heterogeneity ($I^2$: 0%, $\tau^2$: 0), when LVEF was measured with 2DE (S3 Fig). On the other hand, HR for RVEF was 1.29 times larger than that of LVEF with no statistical difference (95% CI: 0.98–1.69, p = 0.067) and moderate heterogeneity ($I^2$: 56.7%, $\tau^2$: 0.049), when LVEF was measured with 3DE (S3 Fig). There was a significant difference in HR ratio between the two groups (p <0.001). When restricted to

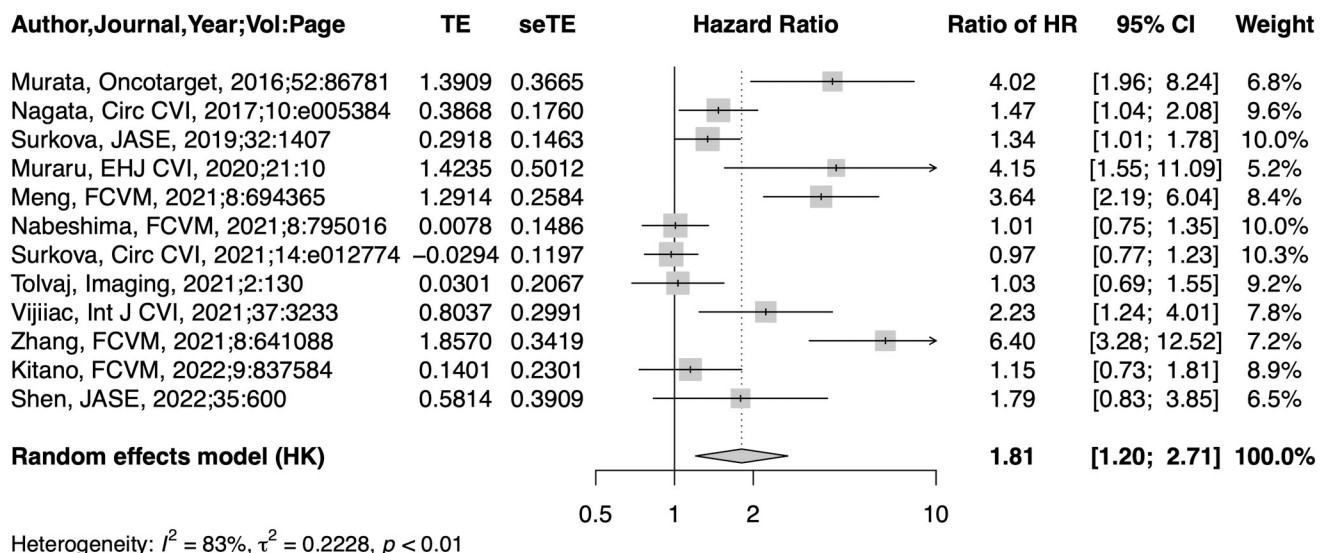

**Fig 3. Forest plots of ratio of HR per 1-SD reduction between RVEF and LVEF.**

studies with reduced LVEF, the pooled ratio of HRs between RVEF and LVEF was 1.34 (95% CI: 0.94–1.91, p = 0.092). (Fig 4).

## Prognostic value of RVEF compared to LVGLS

In nine studies in which both LVGLS and RVEF were presented, a 1-SD reduction in LVGLS (HR: 1.98, 95% CI: 1.60–2.46, p<0.001) or RVEF (HR: 2.20, 95% CI: 1.91–2.52, p<0.001) was significantly associated with death or CV events (S4 Fig). Both LVGLS ($I^2$: 48%, $\tau^2$: 0.028) and RVEF ($I^2$: 46%, $\tau^2$: 0.016) had low heterogeneity. Fig 5 shows forest plots of ratios of HR between RVEF and LVGLS. The pooled ratio of HR was 1.10 (95% CI: 0.91–1.31, p = 0.277). There was low heterogeneity ($I^2$: 8.7%, $\tau^2$: 0.005). LVGLS was measured with 2DE in three studies and with 3DE in six studies. The 95% CI of the HR ratio crossed 1 in both subgroups

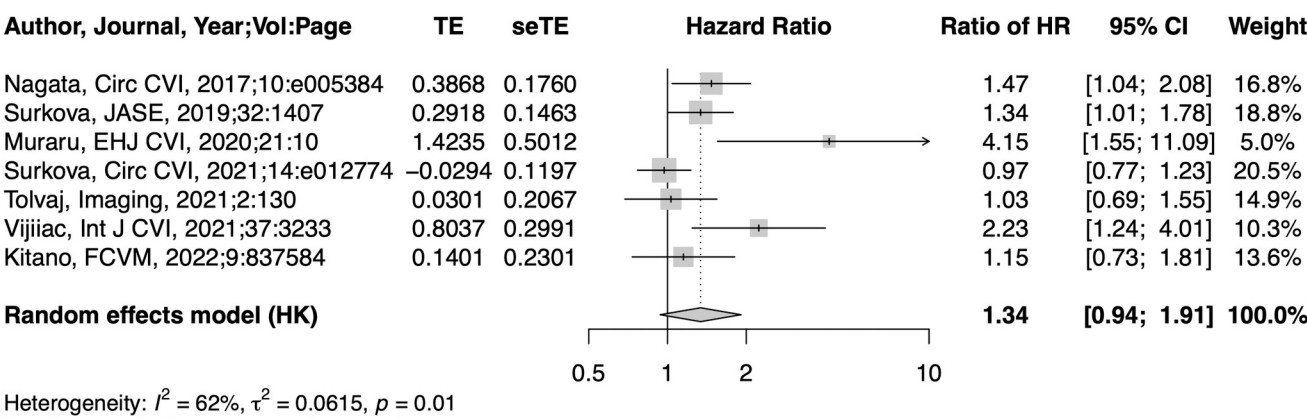

**Fig 4. Forest plots of the ratio of HR per SD reduction between RVEF and LVEF in studies with reduced LVEF.**

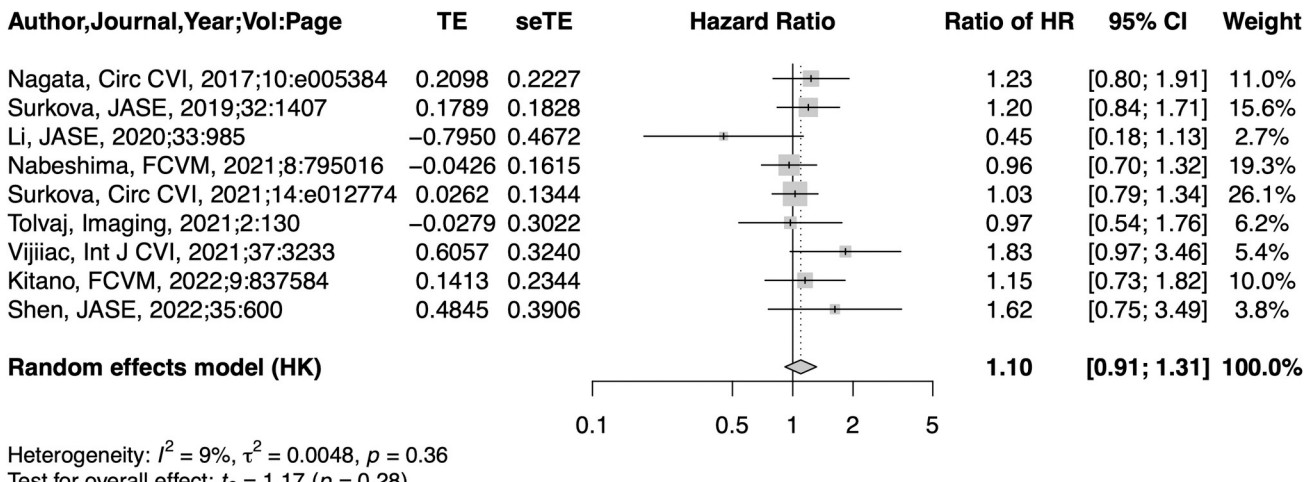

| Author,Journal,Year;Vol:Page | TE | seTE | Hazard Ratio | Ratio of HR | 95% CI | Weight |
|---|---|---|---|---|---|---|
| Nagata, Circ CVI, 2017;10:e005384 | 0.2098 | 0.2227 | | 1.23 | [0.80; 1.91] | 11.0% |
| Surkova, JASE, 2019;32:1407 | 0.1789 | 0.1828 | | 1.20 | [0.84; 1.71] | 15.6% |
| Li, JASE, 2020;33:985 | −0.7950 | 0.4672 | | 0.45 | [0.18; 1.13] | 2.7% |
| Nabeshima, FCVM, 2021;8:795016 | −0.0426 | 0.1615 | | 0.96 | [0.70; 1.32] | 19.3% |
| Surkova, Circ CVI, 2021;14:e012774 | 0.0262 | 0.1344 | | 1.03 | [0.79; 1.34] | 26.1% |
| Tolvaj, Imaging, 2021;2:130 | −0.0279 | 0.3022 | | 0.97 | [0.54; 1.76] | 6.2% |
| Vijiiac, Int J CVI, 2021;37:3233 | 0.6057 | 0.3240 | | 1.83 | [0.97; 3.46] | 5.4% |
| Kitano, FCVM, 2022;9:837584 | 0.1413 | 0.2344 | | 1.15 | [0.73; 1.82] | 10.0% |
| Shen, JASE, 2022;35:600 | 0.4845 | 0.3906 | | 1.62 | [0.75; 3.49] | 3.8% |
| **Random effects model (HK)** | | | | **1.10** | **[0.91; 1.31]** | **100.0%** |

Heterogeneity: $I^2 = 9\%$, $\tau^2 = 0.0048$, $p = 0.36$
Test for overall effect: $t_8 = 1.17$ ($p = 0.28$)

**Fig 5. Forest plots of ratio of HR per 1-SD reduction between RVEF and LV global longitudinal strain.**

(S5 Fig). Subgroup analysis showed there was no significant difference between the two groups (p = 0.51).

## Validation analysis

A total of 1,154 patients with diverse cardiovascular diseases from three publication were included. Clinical and echocardiographic characteristics are shown in Table 3. The primary endpoint was cardiac events, including cardiac death, HF hospitalization, ventricular tachyarrhythmia, or myocardial infarction. During a median follow-up of 29.7 months, 194 patients experienced cardiac events. The Kaplan-Meier survival curve significantly stratified risk by 1) LVEF of 45% and RVEF of 45%, and 2) LVGLS of 16% and RVEF of 45% (Log-rank: p < 0.0001) (Fig 6). Univariable Cox proportional hazard analysis revealed that not only continuous values of LVEF, LVGLS, and RVEF, but also categorical values of LVEF (45% and 50%), LVGLS (15% and 16%), and RVEF (45%) were significantly associated with adverse outcomes (Table 4A). Multivariable analyses showed that RVEF was significantly associated with outcomes after adjusting age, sex, chronic kidney disease (CKD), New York Heart Association functional class (NYHA), and LVEF (p = 0.074) and after adjusting age, sex, CKD, NYHA, and LVGLS (p = 0.015) (Table 4B). Subgroup analyses in patients with LVEF < 45% or ≥ 45% and LVGLS < 16% or ≥ 16% also showed that RVEF was significantly associated with adverse outcomes in all analyses (S4 Table). Incremental value analysis revealed that addition of LVEF (LVGLS) had a significant incremental prognostic value over age, sex, CKD, and NYHA. Further addition of RVEF had a significant incremental prognostic value over age, sex, CKD, NYHA, and LVEF (LVGLS) (S6 Fig).

## Discussion

The major findings of this study are summarized as follows: (1) RVEF assessed with 3DE was significantly associated with adverse outcomes in the overall study population, as well as in patients with PAH; (2) HR per 1-SD reduction in RVEF was a 1.81-fold greater risk of adverse outcomes than with HR per 1-SD reduction of LVEF; (3) This trend was remarkable when LVEF was assessed with 2DE, but not significant when assessed with 3DE; (4) HR per 1-SD reduction in RVEF did not differ from that in LVGLS; (5) Validation analyses revealed that

**Table 3. Clinical and echocardiographic characteristics of validation study (n = 1,154).**

| Variables | |
|---|---|
| Age (year) | 70 ± 14 |
| Gender (male) | 653 (57%) |
| Body surface area (/m$^2$) | 1.56 ± 0.20 |
| Heart rate (bpm) | 68 ± 13 |
| Systolic blood pressure (mmHg) | 139 ± 26 |
| Diastolic blood pressure (mmHg) | 74 ± 14 |
| NYHA (I/II/III/IV) | 740/323/84/5 (64%/28%/7%/1%) |
| Ischemic heart disease | 217 (19%) |
| Dilatated cardiomyopathy | 74 (6%) |
| Hypertrophic cardiomyopathy | 28 (2.5%) |
| Valvular heart disease | 503 (43%) |
| Secondary cardiomyopathy | 169 (15%) |
| Pulmonary hypertension | 21 (2%) |
| Congenital heart disease | 5 (0.5%) |
| Other | 137 (12%) |
| Hypertension | 785 (68%) |
| Diabetes | 359 (31%) |
| Hyperlipidemia | 415 (36%) |
| Coronary artery disease | 332 (29%) |
| Chronic kidney disease | 536 (46%) |
| Atrial fibrillation | 105 (9%) |
| Ca-antagonist | 427 (37%) |
| Beta-blocker | 459 (40%) |
| ACEi or ARB | 731 (64%) |
| Diuretics | 333 (29%) |
| Left ventricular end-diastolic volume (mL) | 139.6 ± 55.4 |
| Left ventricular end-systolic volume (mL) | 78.7 ± 48.4 |
| Left ventricular ejection fraction (%) | 46.7 ± 12.6 |
| Maximum left atrial volume index (mL/m$^2$) | 47.1 ± 20.4 |
| Minimum left atrial volume index (mL/m$^2$) | 30.1 ± 17.6 |
| E/e' | 16.3 ± 8.9 |
| Systolic pulmonary artery pressure (mmHg) | 34.6 ± 11.8 |
| Left ventricular global longitudinal strain (%) | 13.8 ± 4.6 |
| Right ventricular ejection fraction (%) | 48.1 ± 9.5 |

Data are expressed as means ± standard deviations or n (%).

ACEi, angiotensin converting enzyme inhibitor; ARB, angiotensin receptor blocker; Ca, calcium; NYHA, New York Heart Association functional class.

RVEF was significantly associated with outcome after adjusting clinical parameters and LVEF or LVGLS; (6) In studies reporting HRs for both RVEF and LVEF or RVEF and LVGLS in the same cohort, RVEF had 1.8-fold greater prognostic power per 1-SD reduction than LVEF, but had predictive value similar to that of LVGLS and to LVEF in patients with reduced LVEF.

## Previous studies

Several systematic reviews and meta-analyses have been conducted previously on the prognostic value of RVEF assessed with CMR. Dong et al. performed a systematic review and meta-

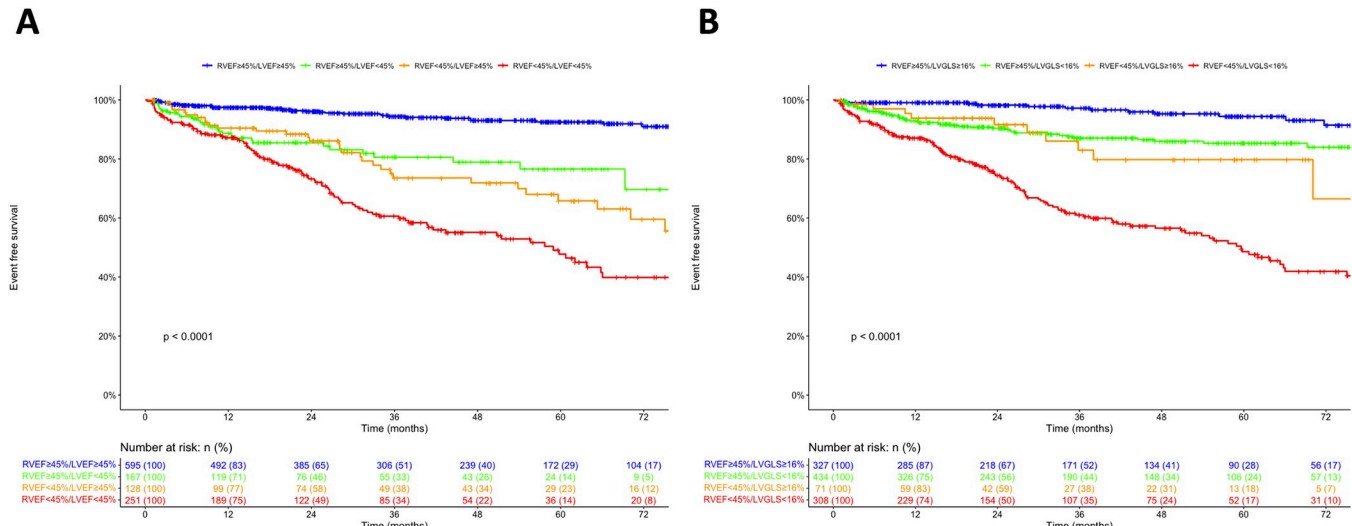

**Fig 6.** Kaplan-Meier survival curves divided into eight groups: (A) LVEF ≥ 45% and RVEF ≥ 45%; LVEF ≥ 45% and RVEF < 45%; LVEF < 45% and RVEF ≥ 45%; LVEF < 45% and RVEF < 45%; (B) LVGLS ≥ 16% and RVEF ≥ 45%; LVGLS ≥ 16% and RVEF < 45%; LVGLS < 16% and RVEF ≥ 45%; LVGLS < 16% and RVEF < 45%. The primary endpoint was cardiac events, including cardiac death, HF hospitalization, ventricular tachyarrhythmia, or myocardial infarction.

analysis to investigate which CMR-derived parameters are associated with adverse outcomes in PAH [29]. Eight articles (1120 patients) used all-cause death as the endpoint and a composite cardiopulmonary event was the endpoint in 10 articles (604 patients). The authors found that only RVEF was a significant prognostic predictor for both all-cause death (pooled HR per 1% increase in RVEF: 0.95, 95% CI: 0.92–0.99, p = 0.014) and a composite endpoint (pooled HR per 1% increase in RVEF: 0.95, 95% CI: 0.93–0.97, p<0.001). Alabed et al. selected 22 studies showing prognostic values of CMR parameters in patients with PAH [30]. Pooled HRs showed that every 1% decrease in RVEF was associated with a 4.9% increase in risk of clinical worsening during 22 months of follow-up and a 2.1% increase in the risk of death over 54 months. In PAH, high pulmonary arterial pressure increases RV diastolic pressure and RV dilatation, leading to decreased RVEF. Hence, it makes sense that RVEF is the strongest prognostic factor for PAH. On the other hand, Papanastasiou et al. performed a systematic review and meta-analysis examining the prognostic value of RVEF in HF [31]. They analyzed 46 articles including 14,344 patients. Meta-analysis revealed that RVEF was a strong predictor of mortality (HR per 10% decrease in RVEF: 1.26, 95% CI: 1.18–1.33) and death or HF hospitalization (HR per 10% decrease in RVEF: 1.31, 95% CI: 1.2–1.42). Results of this study verified that RVEF by CMR provides useful prognostic information even in patients with HF, indicating that it is a robust risk stratification marker in a diverse cardiac disease.

Most recently, a meta-analysis showed that RVEF 3DE is more strongly associated with adverse outcomes than conventional echocardiography RV function parameters [13]. Ten articles (1,928 patients) reporting RVEF and conventional RV systolic parameters (TAPSE, RVFAC, or RVFWLS) on the same cohort were collected and compared for prognostic value. RVEF was robustly associated with adverse outcomes (HR: 2.64 [95% CI: 2.18 to 3.20]) and was significantly and strongly associated with adverse outcomes per 1-SD reduction compared to the other three parameters (vs. TAPSE, HR: 1.54 [95% CI: 1.04 to 2.28]; vs. RVFAC, HR: 1.45 [95% CI: 1.15 to 1.81], vs. RVFWLS, HR: 1.44 [95% CI: 1.07 to 1.95]). However, RVEF has

**Table 4. Univariable and multivariable Cox proportional hazard analysis for cardiac events in validation study.**

A. Univariate analysis

| Variable | N | HR (95% CI) | P value |
|---|---|---|---|
| Age (year) | 1,154 | 1.02 (1.01 to 1.03) | 0.001 |
| Sex (male) | 1,154 | 1.09 (0.82 to 1.45) | 0.5 |
| NYHA (Class II) | 1,154 | 2.04 (1.48 to 2.82) | <0.001 |
| NYHA (Class III/IV) | 1,154 | 5.43 (3.75 to 7.85) | <0.001 |
| CKD | 1,153 | 2.96 (2.19 to 4.00) | <0.001 |
| LVEF (%) | 1,141 | 0.95 (0.94 to 0.96) | <0.001 |
| LVEF < 45% | 1,141 | 4.32 (3.20 to 5.84) | <0.001 |
| LVEF < 50% | 1,141 | 3.81 (2.76 to 5.28) | <0.001 |
| LVGLS (%) | 1,140 | 0.87 (0.84 to 0.89) | <0.001 |
| LVGLS < 15% | 1,140 | 3.88 (2.72 to 5.55) | <0.001 |
| LVGLS < 16% | 1,140 | 4.10 (2.70 to 6.21) | <0.001 |
| RVEF (%) | 1,142 | 0.92 (0.91 to 0.93) | <0.001 |
| RVEF < 45% | 1,142 | 4.95 (3.66 to 6.70) | <0.001 |

B. Multivariable analysis

| Variable | N | Model 1 | | Model 2 | |
|---|---|---|---|---|---|
| | | HR (95% CI) | P value | HR (95% CI) | P value |
| Age (year) | 1,140 | 1.03 (1.01 to 1.04) | <0.001 | 1.03 (1.01 to 1.04) | <0.001 |
| Sex (male) | 1,140 | 0.77 (0.57 to 1.05) | 0.10 | 0.74 (0.54 to 1.01) | 0.055 |
| NYHA (Class II) | 1,140 | 1.22 (0.84 to 1.77) | 0.3 | 1.13 (0.77 to 1.65) | 0.5 |
| NYHA (Class III/IV) | 1,140 | 1.69 (1.07 to 2.68) | 0.024 | 1.58 (0.99 to 2.51) | 0.053 |
| CKD | 1,140 | 1.96 (1.43 to 2.68) | <0.001 | 1.92 (1.40 to 2.64) | <0.001 |
| LVEF (%) | 1,140 | 0.99 (0.97 to 1.00) | 0.074 | | |
| LVGLS (%) | 1,140 | | | 0.95 (0.91 to 0.99) | 0.015 |
| RVEF (%) | 1,140 | 0.94 (0.92 to 0.95) | <0.001 | 0.94 (0.92 to 0.95) | <0.001 |

Cardiac events: Cardiac death, HF hospitalization, ventricular tachyarrhythmia, or myocardial infarction.

CI, confidence interval; CKD, chronic kidney disease; HR, hazard ratio; LVEF, left ventricular ejection fraction; LVGLS, left ventricular global longitudinal strain; NYHA, New York Heart Association functional class; RVEF, right ventricular ejection fraction.

not been compared to LV parameters such as LVEF and LVGLS, which are widely used in the diagnosis, treatment, and prognosis of patients with CV disease.

## Current study

In this study, we systematically surveyed and reviewed literature reporting the prognostic value of RVEF using 3DE. Because we included a diverse group of cardiopulmonary patients with a wide range of RVEF, HRs were re-scaled with the within-study SD to allow direct comparison of HRs across studies. The pooled HR in a 1-SD reduction of RVEF was 2.54 (95% CI: 2.15–3.00, p<0.001), showing that it is a significant predictor of adverse outcomes. The finding was in agreement with the previous meta-analysis [13]. In subgroup analyses among articles, in which study subjects were solely PAH or CV diseases, RVEF was significantly associated with adverse outcomes in both groups, which was consistent with previous meta-analyses in CMR-derived RVEF [29, 30]. We compared the prognostic capacity of RVEF and LVEF or of RVEF and LVGLS by calculating the ratio of HR. The effect on the HR per 1-SD reduction of RVEF was 1.81 times (95% CI: 1.21–2.71, p<0.001) larger than that per 1-SD reduction of LVEF, suggesting superior prognostic value of RVEF compared with LVEF. These results agreed with previous publications showing that the group with reduced RVEF and preserved

LVEF had worse prognosis than the group with preserved RVEF, but reduced LVEF [1, 3]. However, when we only included studies in which LVEF was impaired, the pooled ratio of HRs for RVEF and LVEF crossed 1, indicating there was no superiority of RVEF (Fig 4). We validated the present meta-analysis using individual patient data from our three previous studies and found that RVEF had significant incremental value over LVEF for the association with future prognosis (Fig 6 and S6 Fig). Therefore, loss of statistical significance may be partly related to a decrease in detection power due to a smaller number of studies (n = 7), but further validation is needed. Although the number of articles was small (n = 3), the effect on the HR per 1-SD reduction of RVEF was 4.36 times higher than that in LVEF assessed by 2DE. In contrast, the effect of HR per 1-SD reduction in RVEF was 1.29 times larger than that in LVEF in 9 studies in which LVEF was measured with 3DE, but there were no statistically significant differences (p = 0.067). These results suggest that LVEF measured with 2DE is not robust for predicting future outcomes, possibly due to geometric assumptions, manual editing of the endocardial border, a limited 2DE cutting plane, and observer variabilities.

In contrast, the HR of a 1-SD reduction in RVEF was similar to the HR of a 1-SD reduction in LVGLS in all articles, as well as subgroup analysis based on the method of LVGLS analysis (2DE or 3DE). This suggests that RVEF is as robust as LVGLS in prognostication. LVGLS is obtained with semi-automated software using the speckle-tracking method, resulting in less observer variability. Furthermore, LVGLS is measured from three cross-sectional views, even in 2DE. Therefore, RVEF may be comparable to LVGLS in prognostication.

## Study limitations

There are several limitations of this study. First, as with all meta-analyses, it was influenced by variability of the original studies. Some results showed asymmetry in funnel plots that may have been influenced by publication bias. However, we performed outlier and trim-and-fill analysis, and the observed effect of size was not different. Second, patients with various diseases were analyzed together, which may have influenced these results. However, results of this study are applicable to all patients encountered in daily clinical practice. Third, outcomes and follow-up periods varied across studies, which affected HR. Fourth, the present meta-analysis may include significant bias in the comparison of RVEF and LVEF because many patients with preserved LVEF were included in the study. In this regard, these results should be interpreted with caution, and further validation is warranted. Fifth, the number of included studies was still small; thus, it is not possible to perform extensive meta-regression or subgroup analysis. Sixth, this study was based on meta-analysis, and more detailed results may be obtained when we perform individual participant data meta-analysis. Seventh, in this study, there was some variability among included studies, and the number of studies was still too small for the findings to be conclusive.

## Conclusions

RVEF assessed by 3DE was significantly associated with adverse CV outcomes not only in patients with PAH, but also in those with CV diseases. Therefore, the findings of this meta-analysis highlight and support the use of RVEF assessed by 3DE to predict CV outcomes in routine clinical practice in patients with CV diseases and in those with PAH.

## Supporting information

**S1 Fig.** Forest plots of the hazard ratio (HR) per standard deviation (SD) reduction of right ventricular (RV) ejection fraction (EF) in patients with pulmonary arterial hypertension (upper panel) and cardiovascular disease (lower panel). CI, confidence interval; CV,

cardiovascular; HR, hazard ratio; PAH, pulmonary arterial hypertension; SD, standard deviation; se, standard error; TE, treatment effect.
(PDF)

**S2 Fig.** Forest plots of HR per SD reduction of left ventricular (LV) EF (A) and RVEF (B) in the same cohort from 12 studies. LVEF, left ventricular ejection fraction; RVEF, right ventricular ejection fraction.
(PDF)

**S3 Fig.** Forest plots of the ratio of HR per SD reduction between RVEF and LVEF by two-dimensional echocardiography (2DE) (upper panel) or three-dimensional echocardiography (3DE) (lower panel). 2DE, two-dimensional echocardiography; 3DE, three-dimensional echocardiography.
(PDF)

**S4 Fig.** Forest plots of HR per SD reduction of LV global longitudinal strain (GLS) (A) and RVEF (B) in the same cohort from 9 studies. LVGLS, left ventricular global longitudinal strain.
(PDF)

**S5 Fig.** Forest plots of the ratio of HR per SD reduction between RVEF and LVGLS by 2DE (upper panel) or 3DE (lower panel). GLS, global longitudinal strain.
(PDF)

**S6 Fig. Nested regression models to assess the incremental prognostic value of RVEF for cardiac events.** CKD, chronic kidney disease; NYHA, New York Heart Association functional class.
(PDF)

**S1 Table. Search strategies.**
(PDF)

**S2 Table. Reasons for full-text exclusion.**
(PDF)

**S3 Table. Summary of studies quality analysis.**
(PDF)

**S4 Table.** Multivariable Cox proportional hazards analysis of (A) LVEF < 45%, (B) LVEF $\geq$ 45%, (C) LVGLS < 16%, (D) LVGLS $\geq$ 16% for cardiac events in validation study.
(PDF)

**S1 Data.**
(CSV)

## Author Contributions

**Conceptualization:** Tetsuji Kitano, Masaaki Takeuchi.

**Data curation:** Tetsuji Kitano, Yosuke Nabeshima, Yasufumi Nagata.

**Formal analysis:** Tetsuji Kitano, Yosuke Nabeshima, Masaaki Takeuchi.

**Investigation:** Tetsuji Kitano.

**Methodology:** Tetsuji Kitano, Masaaki Takeuchi.

**Supervision:** Masaaki Takeuchi.

**Validation:** Yasufumi Nagata.

**Writing – original draft:** Tetsuji Kitano, Masaaki Takeuchi.

**Writing – review & editing:** Yasufumi Nagata.

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
