## [Decision Letter · Decision Letter 0]

23 Feb 2023

PONE-D-23-01475Prognostic Value of the Right Ventricular Ejection Fraction using Three-Dimensional Echocardiography: Systematic Review and Meta-AnalysisPLOS ONE

Dear Dr. Kitano,

Thank you for submitting your manuscript to PLOS ONE. After careful consideration, we feel that it has merit but does not fully meet PLOS ONE’s publication criteria as it currently stands. Therefore, we invite you to submit a revised version of the manuscript that addresses the points raised during the review process.

We look forward to receiving your revised manuscript.

Kind regards,

Daniel A. Morris, M.D

Academic Editor

PLOS ONE

Additional Editor Comments:

I would like to thank the authors for submitting this interesting meta-analysis to PlosOne. While the meta-analysis is methodological well performed, there are some limitations concerning the clinical relevance of the findings.

Major Limitations and Comments:

1) Reviewers’ suggestions:

- The reviewers have addressed important comments and suggestions that should be addressed in the meta-analysis.

2) Originality and Further Analyses:

- Recently has been also published in JASE (Sayour et al. JASE 2023) a meta-analysis on the prognostic relevance of 3DRVEF including the same studies, with similar findings, and even showing also the incremental prognostic relevance of 3DRVEF over conventional RV parameters such as TAPSE, FAC, and RV strain. Hence, as this meta-analysis is lack of originality, further analyses should be presented to increase the clinical relevance of the findings. In this respect, the following further analyses should be performed:

I) Validate the incremental prognostic value of 3DRVEF over conventional RV parameters such as TAPSE, FAC, S-TDI, and RV global and free wall strain.

II) Show in two further separate tables, one for PAH and another for “various CVD”, the main clinical characteristics of the studies such as sample size, feasibility of 3DRVEF, value of LVEF (2D or 3D), value of 3DRVEF, primary outcome analysis, follow-up time, and HR with 95%CI in a continuous analysis (i.e. – 1 SD) and in a dichotomous analysis (i.e. 3DRVEF < 45%).

III) Concerning the analysis about the prognostic value of RVEF vs LVEF, exclude of this analysis studies with only LVEF > 50%. In addition, in the studies and analyses comparing RVEF vs LVEF, also comparing the prognostic relevance of RVEF < 45% vs LVEF < 45%.

IV) Concerning the analysis about the prognostic value of RVEF vs GLS, comparing also the prognostic relevance of RVEF < 45% vs GLS < 16%.

V) Perform a validation analysis using individual patient data from the studies of Takeuchi`s research group (PMID: 33153858 and 35282348) concerning the following analyses: feasibility of 3DRVEF, prognostic relevance of 3DRVEF < 45% and < 30% and also comparing 3DRVEF < 45% vs LVEF < 45%.

VI) Highlight in the abstract and also in the results and discussion section that “the current medical evidence is limited on the potential prognostic relevance of 3DRVEF in patients with HFrEF and HFpEF” and “hence, further studies are warrant to determine the clinical usefulness of this potential new RV parameter in patients with HF”.

Reviewers' comments:

Reviewer's Responses to Questions

**Comments to the Author**

1. Is the manuscript technically sound, and do the data support the conclusions?

Reviewer #1: Yes

Reviewer #2: Yes

2. Has the statistical analysis been performed appropriately and rigorously? 

Reviewer #1: Yes

Reviewer #2: Yes

3. Have the authors made all data underlying the findings in their manuscript fully available?

Reviewer #1: Yes

Reviewer #2: Yes

4. Is the manuscript presented in an intelligible fashion and written in standard English?

Reviewer #1: Yes

Reviewer #2: Yes

5. Review Comments to the Author

Reviewer #1: 

Thank you for the opportunity to review the manuscript by Tetsuji Kitano et al.

In this thorough meta-analysis, the authors analyzed fifteen articles with 3,228 subjects. In the included studies there is some variability and the number of studies was small for a meta-analysis. Hence, these limitations should be mentioned.

Reviewer #2: 

This is a very well written manuscript. Although very interesting, I have some comments especially about the design / study population:

- the study population is not clearly identified and though it's interesting to find the RVEF is an overall excellent predictor of outcome, an explanation on the role of RV in "non RV-related" diseases is important.

- it would be preferable to make the distinction within the results - "pulmonary hypertension " - "other CV diseases" - All CVD including PH

6. PLOS authors have the option to publish the peer review history of their article (what does this mean?). If published, this will include your full peer review and any attached files.

Reviewer #1: **Yes: **Aravind Kumar Radha krishnan

Reviewer #2: **Yes: **Prof. Pamela Moceri

---

## [Author Response · Author response to Decision Letter 0]

11 Mar 2023

We are grateful to the editors and the reviewers for their thoughtful and constructive comments regarding our manuscript, “Prognostic Value of the Right Ventricular Ejection Fraction using Three-Dimensional Echocardiography: Systematic Review and Meta-Analysis (PONE-D-23-01475)”. Our point-by-point responses are given below in blue. Red denotes new or revised text in the manuscript.

Additional Editor Comments:

I would like to thank the authors for submitting this interesting meta-analysis to PlosOne. While the meta-analysis is methodological well performed, there are some limitations concerning the clinical relevance of the findings.

RESPONSE: Thank you for your positive and constructive comments.

Major Limitations and Comments:

1) Reviewers’ suggestions:

- The reviewers have addressed important comments and suggestions that should be addressed in the meta-analysis.

RESPONSE: We addressed the reviewer’s comments below.

2) Originality and Further Analyses:

- Recently has been also published in JASE (Sayour et al. JASE 2023) a meta-analysis on the prognostic relevance of 3DRVEF including the same studies, with similar findings, and even showing also the incremental prognostic relevance of 3DRVEF over conventional RV parameters such as TAPSE, FAC, and RV strain. Hence, as this meta-analysis is lack of originality, further analyses should be presented to increase the clinical relevance of the findings. In this respect, the following further analyses should be performed:

RESPONSE: Thank you for your comments. This study compared the prognostic value of three-dimensional (3D) right ventricular (RV) ejection fraction (EF) to left ventricular (LV) EF or LV global longitudinal strain (GLS), whereas the study by Sayour et al. compared that of RVEF to tricuspid annular plane systolic excursion (TAPSE), fractional area change (FAC) or free-wall longitudinal strain (FWLS). In this respect, we believe that the present study provides novelty and clinical relevance.

 In addition, it is described that PLOS ONE editors evaluates research based on "scientific validity," "rigorous methodology," and "high ethical standards," not on whether the research is trendy or important in the eyes of a few editors (Please see below).

 Taking the above into consideration, we would like you to evaluate our research once again.

We incorporated Sayour’s study in the discussion.

Page 12-13, lines 228-237:

Most recently, a meta-analysis showed that RVEF 3DE is more strongly associated with adverse outcomes than conventional echocardiography RV function parameters [15]. Ten articles (1,928 patients) reporting RVEF and conventional RV systolic parameters (TAPSE, RVFAC, or RVFWLS) on the same cohort were collected and compared for prognostic value. RVEF was robustly associated with adverse outcomes (HR: 2.64 [95% CI: 2.18 to 3.20]) and was significantly and strongly associated with adverse outcomes per 1-SD reduction compared to the other three parameters (vs. TAPSE, HR: 1.54 [95% CI: 1.04 to 2.28]; vs. RVFAC, HR: 1. 45 [95% CI: 1.15 to 1.81], vs. RVFWLS, HR: 1.44 [95% CI: 1.07 to 1.95]). However, RVEF by has not been compared to LV parameters such as LVEF or LVGLS.

Page 13, lines 243- 245:

The pooled HR in a 1-SD reduction of RVEF was 2.54 (95% CI: 2.15 - 3.00, p<0.001), showing that it is a significant predictor of adverse outcomes. The finding was in agreement with the previous meta-analysis [15].

I) Validate the incremental prognostic value of 3DRVEF over conventional RV parameters such as TAPSE, FAC, S-TDI, and RV global and free wall strain.

RESPONSE: To begin with, the purpose of this study was (1) to examine the prognostic value of RVEF and (2) to compare its prognostic value with that of LVEF or LVGLS. This study was never performed to demonstrate the validity of the study by Sayour et al.

 This is for your records. Among the 15 eligible articles in our study, 10 were duplicates of the articles included in Sayour et al. Furthermore, four of the other five articles did not investigate hazard ratios (HRs) for TAPSE, FAC, or FWLS. The remaining one article by Li et al. (J Am Soc Echocardiogr. 2020;33:985-994.) reported only the HR of FAC in addition to RVEF.

 Although our aim was not to validate Sayour’s study, addition of HR of FAC from only one article is not enough to validate the study by Sayour et al.

II) Show in two further separate tables, one for PAH and another for “various CVD”, the main clinical characteristics of the studies such as sample size, feasibility of 3DRVEF, value of LVEF (2D or 3D), value of 3DRVEF, primary outcome analysis, follow-up time, and HR with 95%CI in a continuous analysis (i.e. – 1 SD) and in a dichotomous analysis (i.e. 3DRVEF < 45%).

RESPONSE: Thank you for your constructive comments. We have created a table as you indicated (Table 2). 

Page 8, lines 129-136:

Table 2: Main clinical findings of studies with PAH and various CVD.

A: PAH

First author Year Number Feasibility RVEF (%) LVEF (%) LVGLS (%) Primary outcome Follow-up (months) HR by RVEF HR by LVEF HR by LVGLS

Murata 2016 86 0.87 43±12 69±8 NA Composite 14.1 0.92 (0.88-0.96) 1.05 (0.99-1.12) NA

Moceri 2018 104 0.92 35.6±9.7 69.1±8.5 -8.4±3.6 CPD 6.6 NA NA NA

Li 2020 54 0.81 38±10 NA -15.4±6.8 Composite 28 0.92 (0.87-0.98) NA 1.27 (1.15-1.41)

Li 2021 203 0.89 37.5±9.1 NA -18.6±5.5 Composite 20.9 0.90 (0.84-0.95) NA NA

B: Various CVDs

First author Year Number Feasibility RVEF (%) LVEF (%) LVGLS (%) Primary outcome Follow-up (months) HR by RVEF HR by LVEF HR by LVGLS

Nagata 2017 446 0.98 50±10 48±13 -12.8±4.1 CD 49.2 0.90 (0.88-0.93) 0.94 (0.92-0.96) 1.23 (1.13-1.34)

Surkova 2019 394 0.85 47±8.1 53.6±13.3 -16.6±5.3 ACD 44.4 0.90 (0.88-0.92) 0.96 (0.94-0.97) 1.14 (1.07-1.21)

Muraru 2020 412 0.63 47±7.4 53.7±13.3 NA Composite 44.5 0.91 (0.81-0.96) 0.93 (0.81-0.97) NA

Surkova 2021 292 0.85 46.5±9.2 49.5±14.3 -14.7±5.1 Composite 80.4 0.94 (0.93-0.96) 0.96(0.95-0.97) 1.11 (1.07-1.16)

Kitano 2022 341 0.98 46.7±11.1 39.7±16.3 -15.2±4.7 Composite 19.8 0.93 (0.91-0.96) 0.96 (0.94-0.98) 1.12 (1.05-1.19)

Data are expressed as number or means ± standard deviations.

CVD, cardiovascular disease; HR, hazard ratio; LVEF, left ventricular ejection fraction; LVGLS, left ventricular global longitudinal strain; PAH, pulmonary arterial hypertension; RVEF, right ventricular ejection fraction. Other abbreviations are the same as in Table 1. 

III) Concerning the analysis about the prognostic value of RVEF vs LVEF, exclude of this analysis studies with only LVEF > 50%. In addition, in the studies and analyses comparing RVEF vs LVEF, also comparing the prognostic relevance of RVEF < 45% vs LVEF < 45%.

RESPONSE: Since our study is a study level meta-analysis and not an individual participant data meta-analysis, we are sorry, but we cannot perform the analysis as you suggested. In addition, we could not perform the suggested analyses because no other study reported the results of the dichotomous analysis except one study (Front Cardiovasc Med. 9:837584.).

IV) Concerning the analysis about the prognostic value of RVEF vs GLS, comparing also the prognostic relevance of RVEF < 45% vs GLS < 16%.

RESPONSE: Except for one previous study from our laboratory, no other studies have reported the results of dichotomous analysis, so we were not able to perform the analysis that you suggested.

V) Perform a validation analysis using individual patient data from the studies of Takeuchi`s research group (PMID: 33153858 and 35282348) concerning the following analyses: feasibility of 3DRVEF, prognostic relevance of 3DRVEF < 45% and < 30% and also comparing 3DRVEF < 45% vs LVEF < 45%.

RESPONSE: At first, the patients in PMID: 33153858 and PMID: 35282348 articles were overlapped. Therefore, we examined the validity of this study using data from three previous studies reported by our group (Circ Cardiovasc Imaging. 2017;10:e005384., Front Cardiovasc Med. 8:795016., Front Cardiovasc Med. 9:837584.). A total of 1,154 patients from the three studies were included, with a feasibility of 0.98 for RVEF. RVEF <45% had a significant prognostic value, while RVEF <30% did not. RVEF<45% had a comparable hazard ratio to LVEF<45%. These results were consistent with the results of this meta-analysis.

 Hazard ratio 95% CI P value

RVEF<45% 1.32 1.13 – 1.54 0.0007

RVEF<30% 1.30 0.92 – 1.84 0.1346

LVEF<45% 1.35 1.15 – 1.57 0.0002

These results are only provided for the editor. We did not incorporate the results in the manuscript.

VI) Highlight in the abstract and also in the results and discussion section that “the current medical evidence is limited on the potential prognostic relevance of 3DRVEF in patients with HFrEF and HFpEF” and “hence, further studies are warrant to determine the clinical usefulness of this potential new RV parameter in patients with HF”.

RESPONSE: Thank you for the good suggestion. We have added the phrases you indicated in the discussion section.

Page 14, lines 273-275: 

The current medical evidence is limited on the potential prognostic relevance of RVEF in patients with HF. Hence, further studies are warranted to determine the clinical usefulness of this potential new RV parameter in patients with HF. 

Reviewer #1: 

Thank you for the opportunity to review the manuscript by Tetsuji Kitano et al.

In this thorough meta-analysis, the authors analyzed fifteen articles with 3,228 subjects. In the included studies there is some variability and the number of studies was small for a meta-analysis. Hence, these limitations should be mentioned.

RESPONSE: Thank you for the comment. We have described this limitation in the study limitations section.

Page 15, lines 288-289:

Sixth, in this study, there was some variability among the included studies, and the number of studies was still small for obtaining valid conclusions. 

Reviewer #2: 

This is a very well written manuscript. Although very interesting, I have some comments especially about the design / study population:

RESPONSE: Thank you for your positive comments.

- the study population is not clearly identified and though it’s interesting to find the RVEF is an overall excellent predictor of outcome, an explanation on the role of RV in “non RV-related” diseases is important.

RESPONSE: Table 1 summarizes the characteristics of the subjects included in this study. In addition, the main clinical findings for PAH and various CVD are now summarized in Table 2

 We performed a subgroup analysis, dividing PAH as RV-related diseases and various CVD as non-RV-related diseases (S1 Fig). RVEF had a prognostic value even in non-RV disease.

Supporting Information

S1 Fig. Forest plots of hazard ratio (HR) per standard deviation (SD) reduction of right ventricular (RV) ejection fraction (EF) in patients with pulmonary arterial hypertension (upper panel) and cardiovascular disease (lower panel).

- it would be preferable to make the distinction within the results – “pulmonary hypertension” – “other CV diseases” – All CVD including PH

RESPONSE: Thank you. We have corrected the result section as suggested.

Page 9, line 138:

All cardiovascular diseases, including pulmonary arterial hypertension

Page 9-10, lines 156-163:

Pulmonary arterial hypertension

In four studies that dealt with PAH, the pooled estimate was 2.79 (95% CI: 2.04 – 3.81, p<0.001) (S1 Fig). 

Various cardiovascular diseases

In five studies investigating various cardiovascular diseases (CVD), a 1SD reduction of RVEF was associated with a 2.24-fold (95% CI: 1.77 – 2.83, p<0.001) increase in risk for adverse outcomes (S1 Fig). 

and

RESPONSE: We have checked PLOS ONE's style requirements and have corrected our manuscript.

RESPONSE: Confirmed.

RESPONSE: Confirmed.

---

## [Editor Report · Decision Letter 1]

14 Mar 2023

PONE-D-23-01475R1Prognostic value of the right ventricular ejection fraction using three-dimensional echocardiography: Systematic review and meta-analysisPLOS ONE

Dear Dr. Kitano,

Thank you for submitting your manuscript to PLOS ONE. After careful consideration, we feel that it has merit but does not fully meet PLOS ONE’s publication criteria as it currently stands. Therefore, we invite you to submit a revised version of the manuscript by Apr 28 2023 11:59PM that addresses the points raised during the review process.

Please include the following items when submitting your revised manuscript:A rebuttal letter that responds to each point raised by the academic editor and reviewer(s). You should upload this letter as a separate file labeled 'Response to Reviewers'.A marked-up copy of your manuscript that highlights changes made to the original version. You should upload this as a separate file labeled 'Revised Manuscript with Track Changes'.An unmarked version of your revised paper without tracked changes. You should upload this as a separate file labeled 'Manuscript'.

We look forward to receiving your revised manuscript.

Kind regards,

Daniel A. Morris, M.D

Academic Editor

PLOS ONE

Additional Editor Comments:

Thank you very much for your time in revising the manuscript and for submitting your meta-analysis again to PlosOne. While some limitations have been well addressed, there are pending major limitations that should be mandatorily performed to reach acceptance in PlosOne.

Pending Major Limitations and Comments:

1) Validation analysis:

- The further analysis of individual patient data showing and comparing the prognostic relevance of RVEF vs. LVEF in a continuous and dichotomous analysis (i.e., RVEF < 45% vs. LVEF < 45%) using the data of the authors (Takeuchi`s research group) should be shown in a table in the results section and highlighted and discussed in the manuscript as a major finding. In addition, a separate table should show the clinical and echocardiographic characteristics of the population analyzed in this validation analysis.

- Furthermore, also using the available data of your research group (i.e., Takeuchi’s research group), please perform and compare in a second validation analysis the prognostic relevance of RVEF vs. GLS in a continuous and dichotomous analysis (i.e., RVEF < 45% vs. GLS < 16%).

- These validation analyses are the main findings by which the paper will have clinical relevance given: I) the major limitations and bias of comparing RVEF vs LVEF in patients with preserved LVEF; II) the very low number of studies in patients with HFrEF or HFmrEF; and III) the lack of novelty in comparison with the meta-analysis of Sayour et al. JASE 2023 (in effect, one of the major goals of your meta-analysis was to determine the prognostic relevance of 3DRVEF (as stated in the manuscript - page 2 – paragraph 1)).

2) Refine the results by excluding the studies with preserved LVEF (i.e., LVEF ≥ 50%):

- Including patients with preserved LVEF is a major and clinically relevant issue and limitation of the meta-analysis since it is expected and logical that LVEF will not have prognostic relevance in those patients with LVEF ≥ 50% because the values of LVEF are normal. Hence, the authors should refine the results. In this respect, the following studies (all with preserved LVEF) should be mandatorily excluded in the comparison of RVEF vs. LVEF (i.e., the studies of the following authors et al.: Murata, Moceri, Li, Li, Meng, Nabeshima (70% had LVEF ≥ 50%), Zhang, and Shen should be excluded in the comparison of the prognostic relevance of RVEF vs LVEF).

- This major issue should also be stated and discussed in detail in the method and discussion sections, respectively.

3) Further discussion and correction of some important sentences:

- Please correct and discuss with more detail the following sentences: I) “However, the prognostic usefulness of RVEF using 3DE has not been fully investigated systematically” page 3 – paragraph 2; II) “However, RVEF by has not been compared to LV parameters such as LVEF or LVGLS” page 13 – paragraph 1; III) “The current medical evidence is limited on the potential prognostic relevance of RVEF in patients with HF. Hence, further studies are warranted to determine the clinical usefulness of this potential new RV parameter in patients with HF” page 14 – paragraph 4.

---

## [Author Response · Author response to Decision Letter 1]

25 May 2023

We are grateful to for the editor’s thoughtful and constructive comments regarding our manuscript, “Prognostic Value of the Right Ventricular Ejection Fraction using Three-Dimensional Echocardiography: Systematic Review and Meta-Analysis (PONE-D-23-01475R1)”. In accordance with the comments, we performed individual patient data analysis using three publications from our group (Nagata, Circ CVI; Nabeshima, FCVM; Kitano, FCVM). We omitted one publication (Namisaki et al., JASE 2021;34:117-26) because patients were overlapped in the publication in Kitano et al (FCVM 2022;9:837584), and Kitano’s publication had a larger number of patients. We also apologize for the delay required to revise the manuscript because we need another Ethical Review Committee approval for conducting new analyses (UOEHCRB23-023). Our point-by-point responses are given below in blue. Red denotes new or revised text in the manuscript.

Additional Editor Comments:

Thank you very much for your time in revising the manuscript and for submitting your meta-analysis again to PlosOne. While some limitations have been well addressed, there are pending major limitations that should be mandatorily performed to reach acceptance in PlosOne.

Pending Major Limitations and Comments:

1) Validation analysis:

- The further analysis of individual patient data showing and comparing the prognostic relevance of RVEF vs. LVEF in a continuous and dichotomous analysis (i.e., RVEF < 45% vs. LVEF < 45%) using the data of the authors (Takeuchi`s research group) should be shown in a table in the results section and highlighted and discussed in the manuscript as a major finding. In addition, a separate table should show the clinical and echocardiographic characteristics of the population analyzed in this validation analysis.

RESPONSE: We performed a validation analysis using individual patient data from our previous studies according to the comments. Based on the validation analysis, we revised the Materials and Methods, Results, and Discussion sections of the manuscript. The results of the validation analysis were given in Tables 3.

 The analysis using individual patient data included 1,154 patients, with a mean age of 70 ± 14 years (57% male). Left ventricular (LV) ejection fraction (EF), LV global longitudinal strain (GLS), and right ventricular (RV) EF were 46.7 ± 12.6%, 13.8 ± 4.6%, and 48.1 ± 9.5%, respectively. During a median follow-up of 29.7 months, 194 patients had cardiac events. We divided patients into four groups according to 1) LVEF of 45% and RVEF of 45%, 2) LVGLS of 16% and RVEF of 45%, and performed Kaplan-Meyer survival analysis. The results are shown in new figure. We also performed univariable and multivariable Cox proportional hazard analyses. Univariable analysis revealed that not only continuous values of LVEF, LVGLS, and RVEF but also categorical values of LVEF (45% and 50%), GLS (15% and 16%), and RVEF (45%) were significantly associated with adverse outcome. Multivariable analyses showed that RVEF was significantly associated with outcome after adjusting age, sex, CKD, NYHA, and LVEF (p=0.074) and after adjusting age, sex, CKD, NYHA, and LVGLS (p=0.015). We also performed subgroup analyses in patients whose LVEF was <45% or ≥ 45% and those whose GLS was < 16% or ≥ 16%. All analyses showed that RVEF was significantly associated with adverse outcome. Finally, we performed incremental analysis. Addition of LVEF (LVGLS) had a significant incremental value over age, sex, CKD, and NYHA. Further addition of RVEF had a significant incremental value over age, sex, CKD, NYHA, and LVEF (LVGLS). All results confirmed that RVEF is a robust predictor for future outcome. We included these points in this revise manuscript.

Page 6, lines 98-102:

Validation analysis

To validate the results of this meta-analysis, a validation analysis was conducted using individual patient data from three of four previous studies in our laboratory [1, 2, 15]. One study was excluded because majority of patients were overlapped and the number of study population was smaller than the others.

Page 7, lines 124-130:

Regarding validation analysis, we performed Kaplan-Meyer survival analysis among four groups according to 1) LVEF < 45% or ≥ 45% and RVEF < 45% or ≥ 45%, and 2) LVGLS < 16% or ≥ 16% and RVEF < 45% or ≥ 45%. Next, uni- and multi-variable Cox proportional hazards regression analyses were performed. Subgroup analyses were also conducted in patients with LVEF < 45% or > 45% and LVGLS < 16% or > 16%. Finally, sequential Cox regression analysis was conducted to determine incremental value of RVEF over LVEF (LVGLS) and clinical parameters.

Page 13, lines 219-236:

Validation analysis

A total of 1,154 patients with diverse cardiovascular diseases from three publication were included. Clinical and echocardiographic characteristics were shown in Table 3. During a median follow-up of 29.7 months, 194 patients had cardiac events. Kaplan-Meier survival curve significantly stratified risk by 1) LVEF of 45% and RVEF of 45%, and 2) LVGLS of 16% and RVEF of 45% (Log-rank: p < 0.0001) (S7 Fig). Univariable Cox proportional hazard analysis revealed that not only continuous values of LVEF, LVGLS, and RVEF but also categorical values of LVEF (45% and 50%), GLS (15% and 16%), and RVEF (45%) were significantly associated with adverse outcome (Table S4 A). Multivariable analyses showed that RVEF was significantly associated with outcome after adjusting age, sex, chronic kidney disease (CKD), New York Heart Association functional class (NYHA), and LVEF (p=0.074) and after adjusting age, sex, CKD, NYHA, and LVGLS (p=0.015) (Table S4 B). Subgroup analyses in patients with LVEF < 45% or ≥ 45% and GLS < 16% or ≥ 16% also showed that RVEF was significantly associated with adverse outcomes in all analyses (Table S5). Incremental value analysis revealed that addition of LVEF (GLS) had a significant incremental prognostic value over age, sex, CKD, and NYHA. Further addition of RVEF had a significant incremental prognostic value over age, sex, CKD, NYHA, and LVEF (GLS) (S8 Fig).

Page 16, lines 300-306:

However, when we only included studies in which LVEF was impaired, the pooled ratio of HRs for RVEF and LVEF crossed 1, indicating there was no superiority of RVEF (S4 Fig). We validated the present meta-analysis using individual patient data from our three previous studies and found that RVEF had a significant incremental value over LVEF for the association with future prognosis (Table S7, S8 Fig). Therefore, loss of statistical significance may be partly related to a decrease in detection power due to a smaller number of studies (n = 7), but further validation is needed.

- Furthermore, also using the available data of your research group (i.e., Takeuchi’s research group), please perform and compare in a second validation analysis the prognostic relevance of RVEF vs. GLS in a continuous and dichotomous analysis (i.e., RVEF < 45% vs. GLS < 16%). 

RESPONSE: We addressed this point in our response to the comment above.

- These validation analyses are the main findings by which the paper will have clinical relevance given: I) the major limitations and bias of comparing RVEF vs LVEF in patients with preserved LVEF; II) the very low number of studies in patients with HFrEF or HFmrEF; and III) the lack of novelty in comparison with the meta-analysis of Sayour et al. JASE 2023 (in effect, one of the major goals of your meta-analysis was to determine the prognostic relevance of 3DRVEF (as stated in the manuscript - page 2 – paragraph 1)).

RESPONSE: Thank you very much for your constructive and thoughtful comments. We understood the significance of conducting a validation analysis and we agreed with you. We added in the limitations of the manuscript that the present meta-analysis might cause significant bias in the comparison of RVEF and LVEF because many patients with preserved LVEF were included in the study.

Page 18, lines 335-337:

Fourth, the present meta-analysis might cause significant bias in the comparison of RVEF and LVEF because many patients with preserved LVEF were included in the study. In this regard, there is caution in interpreting the results, and further validation is warranted.

2) Refine the results by excluding the studies with preserved LVEF (i.e., LVEF ≥ 50%):

- Including patients with preserved LVEF is a major and clinically relevant issue and limitation of the meta-analysis since it is expected and logical that LVEF will not have prognostic relevance in those patients with LVEF ≥ 50% because the values of LVEF are normal. Hence, the authors should refine the results. In this respect, the following studies (all with preserved LVEF) should be mandatorily excluded in the comparison of RVEF vs. LVEF (i.e., the studies of the following authors et al.: Murata, Moceri, Li, Li, Meng, Nabeshima (70% had LVEF ≥ 50%), Zhang, and Shen should be excluded in the comparison of the prognostic relevance of RVEF vs LVEF). 

RESPONSE: Thank you for your suggestions. We created a forest plot excluding the eight studies in which LVEF was preserved (S4 Fig). The pooled ratio of HRs between RVEF and LVEF was 1.34 (95% CI: 0.94 - 1.91).

 As mentioned in our response to the comment #1, we also examined the prognostic value of LVEF, LVGLS, and RVEF in patients who had LVEF < 45% (n = 417) or in those whose LVGLS had < 16% (n= 741), using individual patient data. In patients with LVEF < 45%, RVEF (HR:0.95, 95% CI: 0.93 – 0.97, p < 0.001) was significantly associated with adverse outcome after adjusting age, sex, CKD, NYHA, and LVEF (HR:1.01, 95% CI: 0.99 – 1.03, p = 0.4). In patients whose LVGLS were < 16%, RVEF (HR:0.94, 95% CI: 0.92 – 0.95, p < 0.001) was significantly associated with adverse outcome after adjusting age, sex, CKD, NYHA, and LVGLS (HR:0.99, 95% CI: 0.94 – 1.05, p = 0.7). Thus, individual patient data analysis showed that RVEF had a superior predictor over LVEF (LVGLS) in patients with impaired LVEF (LVGLS). However, we think additional studies should be required to validate these results. This point mentioned in the study limitation.

Page 18, lines 335-337:

Fourth, the present meta-analysis might cause significant bias in the comparison of RVEF and LVEF because many patients with preserved LVEF were included in the study. In this regard, there is caution in interpreting the results, and further validation is warranted.

- This major issue should also be stated and discussed in detail in the method and discussion sections, respectively.

RESPONSE: We revised the Materials and Methods, Results, and Discussion section, as suggested by the editor.

Page 7, lines 124-130:

Regarding validation analysis, we performed Kaplan-Meyer survival analysis among four groups according to 1) LVEF < 45% or ≥ 45% and RVEF < 45% or ≥ 45%, and 2) LVGLS < 16% or ≥ 16% and RVEF < 45% or ≥ 45%. Next, uni- and multi-variable Cox proportional hazards regression analyses were performed. Subgroup analyses were also conducted in patients with LVEF < 45% or > 45% and LVGLS < 16% or > 16%. Finally, sequential Cox regression analysis was conducted to determine incremental value of RVEF over LVEF (LVGLS) and clinical parameters.

Page 12, lines 200-201:

When restricted to studies with reduced LVEF, the pooled ratio of HRs between RVEF and LVEF was 1.34 (95% CI: 0.94 - 1.91, p=0.092). (S4 Fig).

Page 16, lines 300-306:

However, when we only included studies in which LVEF was impaired, the pooled ratio of HRs for RVEF and LVEF crossed 1, indicating there was no superiority of RVEF (S4 Fig). We validated the present meta-analysis using individual patient data from our three previous studies and found that RVEF had a significant incremental value over LVEF for the association with future prognosis (Table S7, S8 Fig). Therefore, loss of statistical significance may be partly related to a decrease in detection power due to a smaller number of studies (n = 7), but further validation is needed.

3) Further discussion and correction of some important sentences:

- Please correct and discuss with more detail the following sentences: I) “However, the prognostic usefulness of RVEF using 3DE has not been fully investigated systematically” page 3 – paragraph 2; II) “However, RVEF by has not been compared to LV parameters such as LVEF or LVGLS” page 13 – paragraph 1; III) “The current medical evidence is limited on the potential prognostic relevance of RVEF in patients with HF. Hence, further studies are warranted to determine the clinical usefulness of this potential new RV parameter in patients with HF” page 14 – paragraph 4.

RESPONSE: We revised the manuscript

Page 4, lines 58-66:

Just recently, Sayour et al. conducted a systematic review and meta-analysis of the association of 3DE derived RVEF with adverse cardiopulmonary outcomes and compared the prognostic value of RVEF to TAPSE, RVFAC, and RVFWLS [13]. However, the prognostic usefulness of RVEF using 3DE has not been fully investigated systematically and compared with LV parameters such as LV ejection fraction (LVEF) and LV global longitudinal strain (LVGLS). However, RVEF has not been compared to LV parameters such as LVEF and LV global longitudinal strain (LVGLS), which are the most important parameters in patients with cardiovascular disease.

Page 15-16, lines 280-283:

However, RVEF by has not been compared to LV parameters such as LVEF or LVGLS. However, RVEF has not been compared to LV parameters such as LVEF and LVGLS, which play pivotal roles in the diagnosis, treatment, and prognosis of patients with cardiovascular disease.

Page 17, lines 321-325:

The current medical evidence is limited on the potential prognostic relevance of RVEF, especially in HF patients with reduced EF. Although validation study verified that RVEF had a better predictor than LVEF or LVGLS in patients with impaired LVEF (LVEF < 45%), further meta-analyses aimed to HF patients with reduced EF are warranted to determine the clinical usefulness of 3DE determined RVEF.

---

## [Editor Report · Decision Letter 2]

7 Jun 2023

PONE-D-23-01475R2Prognostic value of the right ventricular ejection fraction using three-dimensional echocardiography: Systematic review and meta-analysisPLOS ONE

Dear Dr. Kitano,

Thank you for submitting your manuscript to PLOS ONE. After careful consideration, we feel that it has merit but does not fully meet PLOS ONE’s publication criteria as it currently stands. Therefore, we invite you to submit a revised version of the manuscript that addresses the points raised during the review process.

We look forward to receiving your revised manuscript.

Kind regards,

Daniel A. Morris, M.D

Academic Editor

PLOS ONE

Journal Requirements:

Additional Editor Comments:

Thank you very much for your time and efforts in preparing and submitting your revised manuscript. Your meta-analysis, including a large and excellent validation analysis, has significantly improved and is almost ready for publication. In this regard, only minor changes remain to get the final version of this interesting and clinically relevant meta-analysis.

Minor Changes:

- Please correct this sentence in the abstract “In subgroup analysis, HR was significantly associated with…”. Maybe you wanted to write “In subgroup analysis, RVEF was significantly associated with…”

- Please edit and change this pivotal sentence in the abstract “In individual patient data analysis (n = 1,154), RVEF was significantly associated with outcome after adjusting clinical variables and LVEF or LVGLS” as “In individual patient data analysis (n = 1,142), RVEF < 45% was significantly associated with worse CV outcomes (HR 4.95 [95%CI 3.66 - 6.70]), even in patients with reduced or preserved LVEF”.

- Please specify and define in the abstract, in the results section, and in each figure and table what outcomes were analyzed. By the way, an example may be “CV outcomes (HF hospitalization, mortality for HF, or CV mortality)” or “clinical and CV outcomes (mortality for all causes, HF hospitalization, mortality for HF, or CV mortality)”.

- Please change in the abstract and in the whole manuscript the sentence/subtitle “various cardiovascular diseases” to “in patients with CV diseases”

- Please edit and change the conclusion of the abstract to: “The findings of this meta-analysis highlight and support the use of RVEF by means of 3D transthoracic echocardiography to predict CV outcomes in routine clinical practice in patients with CV diseases and in those with PAH”.

- Please edit and change the conclusion in the manuscript to: “RVEF assessed by 3DE was significantly associated with adverse CV outcomes not only in patients with PAH, but also in those with CV diseases. Therefore, the findings of this meta-analysis highlight and support the use of RVEF by means of 3D transthoracic echocardiography to predict CV outcomes in routine clinical practice in patients with CV diseases and in those with PAH”.

- The supplemental figure 4 shows clearly that RVEF has a similar prognostic value to LVEF in studies including also patients with reduced LVEF. Likewise, supplemental table 4 also clearly shows that a reduced RVEF has similar prognostic relevance to a reduced LVEF. Hence, please state in the whole manuscript and in the abstract that “by analyzing studies including patients with reduced LVEF, RVEF had similar usefulness to LVEF to predict CV outcomes”.

- Please include in the main manuscript the excellent and clinically relevant supplemental figures s4 and s7 and the supplemental table s4.

- In order to ensure international patient data protection and to avoid misuse of your outstanding data, please do not include in the data supplement the Excel table “supporting information / Validation analysis”.

---

## [Author Response · Author response to Decision Letter 2]

10 Jun 2023

We appreciate the editor's thoughtful and constructive comments on our manuscript, “Prognostic Value of the Right Ventricular Ejection Fraction using Three-Dimensional Echocardiography: Systematic Review and Meta-Analysis (PONE-D-23-01475R2)”. Our point-by-point responses are given below in blue. Red denotes new or revised text in the manuscript.

Journal Requirements:

RESPONSE: We have checked the list of references.

Additional Editor Comments:

Thank you very much for your time and efforts in preparing and submitting your revised manuscript. Your meta-analysis, including a large and excellent validation analysis, has significantly improved and is almost ready for publication. In this regard, only minor changes remain to get the final version of this interesting and clinically relevant meta-analysis.

RESPONSE: Thank you for your positive comments.

Minor Changes:

- Please correct this sentence in the abstract “In subgroup analysis, HR was significantly associated with…”. Maybe you wanted to write “In subgroup analysis, RVEF was significantly associated with…”

RESPONSE: Thank you for the comments. We have corrected the sentence as suggested.

Page 2, lines 27-29:

In subgroup analysis, RVEF was significantly associated with outcome in pulmonary arterial hypertension (PAH) (HR: 2.79, 95% CI: 2.04-3.82) and various cardiovascular (CV) diseases (HR: 2.23, 95%CI: 1.76-2.83).

- Please edit and change this pivotal sentence in the abstract “In individual patient data analysis (n = 1,154), RVEF was significantly associated with outcome after adjusting clinical variables and LVEF or LVGLS” as “In individual patient data analysis (n = 1,142), RVEF < 45% was significantly associated with worse CV outcomes (HR 4.95 [95%CI 3.66 - 6.70]), even in patients with reduced or preserved LVEF”.

RESPONSE: We have corrected the sentence as suggested.

Page 2, lines 32-35:

In individual patient data analysis (n = 1,142), RVEF < 45% was significantly associated with worse CV outcome (HR: 4.95, 95% CI: 3.66-6.70) after adjusting clinical variables and LVEF or LVGLS, even in patients with reduced or preserved LVEF.

- Please specify and define in the abstract, in the results section, and in each figure and table what outcomes were analyzed. By the way, an example may be “CV outcomes (HF hospitalization, mortality for HF, or CV mortality)” or “clinical and CV outcomes (mortality for all causes, HF hospitalization, mortality for HF, or CV mortality)”.

RESPONSE: We have corrected as suggested.

Page 8, lines 141-144:

The primary endpoint was death (any cause, cardiac or cardiopulmonary death) in five studies and 10 other studies used a composite endpoint [all cause death, cardiac death, heart failure (HF) hospitalization, ventricular tachyarrhythmia, non-fatal myocardial infarction, PAH-related hospitalization, or PAH-related intervention].

Page 9, lines 152-153:

Composite endpoint: all cause death, cardiac death, heart failure hospitalization, ventricular tachyarrhythmia, non-fatal myocardial infarction, PAH-related hospitalization, or PAH-related intervention.

Page 10, lines 163-164:

Composite endpoint: all cause death, cardiac death, heart failure hospitalization, ventricular tachyarrhythmia, non-fatal myocardial infarction, PAH-related hospitalization, or PAH-related intervention.

Page 14, lines 231-233:

The primary endpoint was cardiac events, including cardiac death, HF hospitalization, ventricular tachyarrhythmia, or myocardial infarction.

Page 16, lines 257-258:

The primary endpoint was cardiac events, including cardiac death, HF hospitalization, ventricular tachyarrhythmia, or myocardial infarction.

Page 28, lines 519-521:

Table S4. Multivariable Cox proportional hazards analysis of (A) LVEF < 45%, (B) LVEF ≥ 45%, (C) LVGLS < 16%, (D) LVGLS ≥ 16% for cardiac events in validation study.

Table S4:

Table S4: Multivariate Cox proportional hazards analysis of (A) LVEF < 45%, (B) LVEF ≥ 45%, (C) LVGLS < 16%, (D) LVGLS ≥ 16% for cardiac events in validation study.

Cardiac events: Cardiac death, HF hospitalization, ventricular tachyarrhythmia, or myocardial infarction.

- Please change in the abstract and in the whole manuscript the sentence/subtitle “various cardiovascular diseases” to “in patients with CV diseases”

RESPONSE: We have corrected as suggested.

Page 2, lines 27-29:

In subgroup analysis, RVEF was significantly associated with outcome in pulmonary arterial hypertension (PAH) (HR: 2.79, 95% CI: 2.04-3.82) and various cardiovascular (CV) diseases (HR: 2.23, 95%CI: 1.76-2.83).

Page 7, lines 122-124:

Subgroup analysis was conducted according to the background disease [PAH or various cardiac cardiovascular (CV) disease] and modality (2DE or 3DE).

Page 10, lines 158:

Table 2: Main clinical findings of studies with PAH and various CV diseases.

Page 10, lines 161:

B: Various CV diseases

Page 11, lines 165-166:

CVD, cardiovascular disease; HR, hazard ratio; LVEF, left ventricular ejection fraction; LVGLS, left ventricular global longitudinal strain; PAH, pulmonary arterial hypertension; RVEF, right ventricular ejection fraction.

Page 12, lines 190:

Various cardiovascular CV diseases

Page 13, lines 191-193:

In five studies investigating various cardiovascular CV diseases (CVD), a 1-SD reduction of RVEF was associated with a 2.23-fold (95% CI: 1.76 – 2.83, p<0.001) increase in risk for adverse outcomes (S1 Fig).

Page 19, lines 324-327:

In subgroup analyses among articles, in which study subjects were solely PAH or various CVDs CV diseases, RVEF was significantly associated with adverse outcomes in both groups, which was consistent with previous meta-analyses in CMR-derived RVEF [20, 21].

Page 27, lines 489-490:

CI, confidence interval; CVD, cardiovascular disease; HR, hazard ratio; PAH, pulmonary arterial hypertension; SD, standard deviation; se, standard error; TE, treatment effect.

- Please edit and change the conclusion of the abstract to: “The findings of this meta-analysis highlight and support the use of RVEF by means of 3D transthoracic echocardiography to predict CV outcomes in routine clinical practice in patients with CV diseases and in those with PAH”.

RESPONSE: We have corrected the conclusion of the abstract as suggested.

Page 3, lines 37-41:

Conclusions: RVEF assessed by 3DE is significantly associated with outcome not only in PAH, but also in diverse diseases. RVEF has superior prognostic value over LVEF and similar value to LVGLS. The findings of this meta-analysis highlight and support the use of RVEF assessed by 3DE to predict CV outcomes in routine clinical practice in patients with CV diseases and in those with PAH.

- Please edit and change the conclusion in the manuscript to: “RVEF assessed by 3DE was significantly associated with adverse CV outcomes not only in patients with PAH, but also in those with CV diseases. Therefore, the findings of this meta-analysis highlight and support the use of RVEF by means of 3D transthoracic echocardiography to predict CV outcomes in routine clinical practice in patients with CV diseases and in those with PAH”.

RESPONSE: We have corrected the conclusion as suggested.

Page 21-22, lines 379-386:

RVEF assessed by 3DE was significantly associated with adverse CV outcomes not only in patients with PAH, but also in those with CV diseases. Furthermore, RVEF provided superior prognostic value over LVEF and similar value to LVGLS. Further studies should be performed to investigate whether a combination of both LVGLS and RVEF assessed by 3DE could effectively stratify patients with low, moderate, and high risk for future adverse outcomes. Therefore, the findings of this meta-analysis highlight and support the use of RVEF assessed by 3DE to predict CV outcomes in routine clinical practice in patients with CV diseases and in those with PAH.

- The supplemental figure 4 shows clearly that RVEF has a similar prognostic value to LVEF in studies including also patients with reduced LVEF. Likewise, supplemental table 4 also clearly shows that a reduced RVEF has similar prognostic relevance to a reduced LVEF. Hence, please state in the whole manuscript and in the abstract that “by analyzing studies including patients with reduced LVEF, RVEF had similar usefulness to LVEF to predict CV outcomes”.

RESPONSE: We have added the sentence in the manuscript.

Page 20-21, lines 353-357:

Although validation study verified that RVEF had a better predictor than LVEF or LVGLS in patients with impaired LVEF (LVEF < 45%) by analyzing studies including patients with reduced LVEF, further meta-analyses focusing on HF patients with reduced EF are warranted to determine the clinical usefulness of 3DE-determined RVEF.

- Please include in the main manuscript the excellent and clinically relevant supplemental figures s4 and s7 and the supplemental table s4.

RESPONSE: We have included the S4 Fig, S7 Fig and Table S4 in the main manuscript from the supporting files.

- In order to ensure international patient data protection and to avoid misuse of your outstanding data, please do not include in the data supplement the Excel table “supporting information / Validation analysis”.

RESPONSE: Thank you for the comment. We removed the Excel file.

---

## [Editor Report · Decision Letter 3]

14 Jun 2023

PONE-D-23-01475R3Prognostic value of the right ventricular ejection fraction using three-dimensional echocardiography: Systematic review and meta-analysisPLOS ONE

Dear Dr. Kitano, Thank you for submitting your manuscript to PLOS ONE. After careful consideration, we feel that it has merit but does not fully meet PLOS ONE’s publication criteria as it currently stands. Therefore, we invite you to submit a revised version of the manuscript that addresses the points raised during the review process.

Thank you very much again for your time and efforts in preparing and submitting your revised manuscript. Your meta-analysis has significantly improved and is almost ready for publication. In this regard, only minor changes remain to get the final version of this clinically relevant meta-analysis. Please submit your revised manuscript by Jul 29 2023 11:59PM. If you will need more time than this to complete your revisions, please reply to this message or contact the journal office at plosone@plos.org. Please include the following items when submitting your revised manuscript:A rebuttal letter that responds to each point raised by the academic editor and reviewer(s). You should upload this letter as a separate file labeled 'Response to Reviewers'.A marked-up copy of your manuscript that highlights changes made to the original version. You should upload this as a separate file labeled 'Revised Manuscript with Track Changes'.An unmarked version of your revised paper without tracked changes. You should upload this as a separate file labeled 'Manuscript'.

We look forward to receiving your revised manuscript.

Kind regards,

Daniel A. Morris, M.D

Academic Editor

PLOS ONE

Journal Requirements:

Additional Editor Comments:

Thank you very much again for your time and efforts in preparing and submitting your revised manuscript. Your meta-analysis has significantly improved and is almost ready for publication. In this regard, only minor changes remain to get the final version of this clinically relevant meta-analysis.

Pending Minor Changes:

1) Please correct the sentence in the abstract “In studies reporting HRs for both RVEF and LVEF or RVEF and LVGLS…” as “In studies reporting HRs for both RVEF and LVEF or RVEF and LVGLS in the same cohort, RVEF had 1.8-fold greater prognostic power per 1-SD reduction than LVEF (ratio of HR: 1.81, 95%CI: 1.20-2.71), but had predictive value similar to that of LVGLS (ratio of HR: 1.10, 95%CI: 0.91-1.31) and to LVEF in patients with reduced LVEF (ratio of HR: 1.34, 95%CI: 0.94 - 1.91).

2) Please add in the discussion section in the first paragraph one 6th-point: “ 6) In studies reporting HRs for both RVEF and LVEF or RVEF and LVGLS in the same cohort, RVEF had 1.8-fold greater prognostic power per 1-SD reduction than LVEF, but had predictive value similar to that of LVGLS and to LVEF in patients with reduced LVEF.

3) Please delete the following sentences in the discussion section because these are out of context and contradictories to the findings of the meta-analysis:

- “To our knowledge, this is the first meta-analysis to examine the prognostic value of RVEF using 3DE and comparing prognostic strength of RVEF with that of LVEF or LVGLS”

- “The current medical evidence is limited on the potential prognostic relevance of RVEF, especially in HF patients with reduced EF. Although validation study verified that RVEF had a better predictor than LVEF or LVGLS in patients with impaired LVEF (LVEF < 45%) by analyzing studies including patients with reduced LVEF, further meta-analyses focusing on HF patients with reduced EF are warranted to determine the clinical usefulness of 3DE-determined RVEF.”

4) Please correct the figure legend of figure 6 as: Fig 6. Kaplan-Meier survival curves divided into eight groups: (A) LVEF ≥ 45% and RVEF ≥ 45%; LVEF ≥ 45% and RVEF < 45%; LVEF < 45% and RVEF ≥ 45%; LVEF < 45% and RVEF < 45%; (B) LVGLS ≥ 16% and RVEF ≥ 45%; LVGLS ≥ 16% and RVEF < 45%; LVGLS < 16% and RVEF ≥ 45%; LVGLS < 16% and RVEF < 45%.

---

## [Author Response · Author response to Decision Letter 3]

14 Jun 2023

We thank the editor for their careful and constructive comments on our manuscript, “Prognostic Value of the Right Ventricular Ejection Fraction using Three-Dimensional Echocardiography: Systematic Review and Meta-Analysis (PONE-D-23-01475R3)”. Our point-by-point responses are given below in blue. Red denotes new or revised text in the manuscript.

Journal Requirements:

RESPONSE: We have checked the list of references. We did not include articles that have been retracted.

Additional Editor Comments:

Thank you very much again for your time and efforts in preparing and submitting your revised manuscript. Your meta-analysis has significantly improved and is almost ready for publication. In this regard, only minor changes remain to get the final version of this clinically relevant meta-analysis.

RESPONSE: Thank you for your positive comments.

Pending Minor Changes:

1) Please correct the sentence in the abstract “In studies reporting HRs for both RVEF and LVEF or RVEF and LVGLS…” as “In studies reporting HRs for both RVEF and LVEF or RVEF and LVGLS in the same cohort, RVEF had 1.8-fold greater prognostic power per 1-SD reduction than LVEF (ratio of HR: 1.81, 95%CI: 1.20-2.71), but had predictive value similar to that of LVGLS (ratio of HR: 1.10, 95%CI: 0.91-1.31) and to LVEF in patients with reduced LVEF (ratio of HR: 1.34, 95%CI: 0.94 - 1.91).

RESPONSE: Thank you for pointing this out. We have revised the abstract as suggested.

Page 2, lines 29-33:

In studies reporting HRs for both RVEF and LVEF or RVEF and LVGLS in the same cohort, RVEF had 1.8-fold greater prognostic power per 1-SD reduction than LVEF (ratio of HR: 1.81, 95%CI: 1.20-2.71), but had predictive value similar to that of LVGLS (ratio of HR: 1.10, 95%CI: 0.91-1.31) and to LVEF in patients with reduced LVEF (ratio of HR: 1.34, 95%CI: 0.94 - 1.91).

2) Please add in the discussion section in the first paragraph one 6th-point: “ 6) In studies reporting HRs for both RVEF and LVEF or RVEF and LVGLS in the same cohort, RVEF had 1.8-fold greater prognostic power per 1-SD reduction than LVEF, but had predictive value similar to that of LVGLS and to LVEF in patients with reduced LVEF.

RESPONSE: We have revised the discussion as suggested.

Page 17-18, lines 276-279:

(6) In studies reporting HRs for both RVEF and LVEF or RVEF and LVGLS in the same cohort, RVEF had 1.8-fold greater prognostic power per 1-SD reduction than LVEF, but had predictive value similar to that of LVGLS and to LVEF in patients with reduced LVEF.

3) Please delete the following sentences in the discussion section because these are out of context and contradictories to the findings of the meta-analysis:

- “To our knowledge, this is the first meta-analysis to examine the prognostic value of RVEF using 3DE and comparing prognostic strength of RVEF with that of LVEF or LVGLS”

- “The current medical evidence is limited on the potential prognostic relevance of RVEF, especially in HF patients with reduced EF. Although validation study verified that RVEF had a better predictor than LVEF or LVGLS in patients with impaired LVEF (LVEF < 45%) by analyzing studies including patients with reduced LVEF, further meta-analyses focusing on HF patients with reduced EF are warranted to determine the clinical usefulness of 3DE-determined RVEF.”

RESPONSE: We have revised the discussion as suggested.

Page 18, lines 279-281:

To our knowledge, this is the first meta-analysis to examine the prognostic value of RVEF using 3DE and comparing prognostic strength of RVEF with that of LVEF or LVGLS.

Page 21, lines 351-356:

The current medical evidence is limited on the potential prognostic relevance of RVEF, especially in HF patients with reduced EF. Although validation study verified that RVEF had a better predictor than LVEF or LVGLS in patients with impaired LVEF (LVEF < 45%) by analyzing studies including patients with reduced LVEF, further meta-analyses focusing on HF patients with reduced EF are warranted to determine the clinical usefulness of 3DE-determined RVEF.

4) Please correct the figure legend of figure 6 as: Fig 6. Kaplan-Meier survival curves divided into eight groups: (A) LVEF ≥ 45% and RVEF ≥ 45%; LVEF ≥ 45% and RVEF < 45%; LVEF < 45% and RVEF ≥ 45%; LVEF < 45% and RVEF < 45%; (B) LVGLS ≥ 16% and RVEF ≥ 45%; LVGLS ≥ 16% and RVEF < 45%; LVGLS < 16% and RVEF ≥ 45%; LVGLS < 16% and RVEF < 45%.

RESPONSE: We have revised the figure legend as suggested.

Page 16, lines 250-253:

Fig 6. Kaplan-Meier survival curves divided into eight groups: (A) LVEF ≥ 45% and RVEF ≥ 45%; LVEF ≥ 45% and RVEF < 45%; LVEF < 45% and RVEF ≥ 45%; LVEF < 45% and RVEF < 45%; (B) LVGLS ≥ 16% and RVEF ≥ 45%; LVGLS ≥ 16% and RVEF < 45%; LVGLS < 16% and RVEF ≥ 45%; LVGLS < 16% and RVEF < 45%.

---

## [Editor Report · Decision Letter 4]

15 Jun 2023

Prognostic value of the right ventricular ejection fraction using three-dimensional echocardiography: Systematic review and meta-analysis

PONE-D-23-01475R4

Dear Dr. Kitano,

We’re pleased to inform you that your manuscript has been judged scientifically suitable for publication and will be formally accepted for publication once it meets all outstanding technical requirements.

Kind regards,

Daniel A. Morris, M.D

Academic Editor

PLOS ONE

---

## [Editor Report · Acceptance letter]

27 Jun 2023

PONE-D-23-01475R4 

Prognostic value of the right ventricular ejection fraction using three-dimensional echocardiography: Systematic review and meta-analysis 

Dear Dr. Kitano:

I'm pleased to inform you that your manuscript has been deemed suitable for publication in PLOS ONE. Congratulations! Your manuscript is now with our production department. 

Kind regards, 

on behalf of

Dr. Daniel A. Morris 

Academic Editor

PLOS ONE